

# Significant source of secondary aerosol: formation from gasoline evaporation emissions in the presence of $SO_2$ and $NH_3$

**Tianzeng Chen[1, 3, a], Yongchun Liu[2, a], Qingxin Ma[1, 3, 4, *], Biwu Chu[1, 3, 4], Peng Zhang[1],**

**Changgeng Liu[1], Jun Liu[1, 3], Hong He[1, 3, 4, *]**

[1] State Key Joint Laboratory of Environment Simulation and Pollution Control, Research Center for

Eco-Environmental Sciences, Chinese Academy of Sciences, Beijing 100085, China

[2] Beijing Advanced Innovation center for Soft Matter Science and Engineering, Beijing University

of Chemical Technology, Beijing 100029, China

[3] University of Chinese Academy of Sciences, Beijing 100049, China

[4] Center for Excellence in Regional Atmospheric Environment, Institute of Urban Environment,

Chinese Academy of Sciences, Xiamen 361021, China

[a] These authors contributed equally to this work and should be considered as co-first authors

*Corresponding authors:* qxma@rcees.ac.cn (Qingxin Ma), and honghe@rcees.ac.cn (Hong He)





## Abstract

Gasoline evaporation emissions have become an important anthropogenic source of urban
atmospheric VOCs and secondary organic aerosol (SOA). These emissions have a significant impact
on regional air quality, especially in China where car ownership is growing rapidly. However, the
contribution of evaporation emissions on the secondary aerosol (SA) is not clear in air pollution
complex in which high concentration of $SO_2$ and $NH_3$ was present. In this study, the effects of $SO_2$
and $NH_3$ on SA formation from unburned gasoline vapors were investigated in a 30 $m^3$ indoor smog
chamber. It was found that increase in $SO_2$ and $NH_3$ concentrations could promote linearly the
formation of SA, which could be enhanced by a factor of 1.6−2.6 and 2.0−2.5, respectively. Sulfate
was most sensitive to the $SO_2$ concentration, followed by organic aerosol, which was due not only
to the well-known acid catalytic effect, but also related to the formation of organic sulfur-containing
compounds. In the case of increasing $NH_3$ concentration, ammonium nitrate increased more
significantly than organic aerosol, and nitrogen-containing organics were also enhanced, as revealed
by the results of positive matrix factorization (PMF) analysis. Meanwhile, new particle formation
(NPF) and particle size growth were significantly enhanced in the presence of $SO_2$ and $NH_3$. This
work indicates that gasoline evaporation emissions will be a significant source of SA, especially in
the presence of high concentrations of $SO_2$ and $NH_3$. Meanwhile, these emissions might also be a
potential source of sulfur- and nitrogen-containing organics. Our work provides a scientific basis
for the synergistic emission reduction of secondary aerosol precursors, including $NO_x$, $SO_2$, $NH_3$
and particularly VOCs, to mitigate PM pollution in China.

## Keywords

Secondary inorganic aerosol; Secondary organic aerosol; Sulfur dioxide; Ammonia; Sulfur-



containing organics; Nitrogen-containing organics



## 1 Introduction

Many areas in China such as the Beijing - Tianjin - Hebei region (BTH), Yangtze River Delta (YRD),
Sichuan Basin and Pearl River Delta (PRD) are suffering from severe haze events (Li et al., 2017; Sun et al.,
2016; Shen et al., 2015; He et al., 2014; Huang et al., 2014; Guo et al., 2014; Tan et al., 2009). Haze pollution
has attracted widespread attention in recent years because of its adverse effects on human health, climate
change and visibility (Thalman et al., 2017; Davidson et al., 2005; Pöschl, 2005).
During the haze events, high concentrations of $SO_2$, $NH_3$, and volatile organic compounds (VOCs) have
always been observed (Zou et al., 2015; Liu et al., 2013; Meng et al., 2011; Yang et al., 2009), which are the
precursors of secondary aerosol. Although the emission of $SO_2$ has decreased continuously since 2005 (Lu
et al., 2010), China is still the largest contributor of $SO_2$ emissions in the world, mainly owing to the great
demand for coal combustion (Bauduin et al., 2016). Also, high concentrations of $SO_2$ of more than 100 ppb
(parts per billion) have been observed in northern China, especially during the heating period (Hou et al.,
2016; Tong et al., 2016; Yang et al., 2009). As for atmospheric $NH_3$, as an alkaline inorganic gas, its main
emission source is agricultural practices in China (Zhang et al., 2018; Fu et al., 2015). It has also been
reported that vehicle exhaust also contributes to $NH_3$ emission in the urban areas (Sun et al., 2017).
Sometimes, high concentrations of $NH_3$ of up to 100 ppb have been observed in Beijing, China (Ianniello et
al., 2010). With respect to VOCs, it is well known that aromatics from anthropogenic sources (especially
vehicle-related sources in urban areas) are critical secondary organic aerosol (SOA) precursors (Liu et al.,
2015a; Gordon et al., 2014; Platt et al., 2013; Calvert et al., 2002). These aromatics could react with oxidants
(e.g., $O_3$, OH, and $NO_3$ radicals), and undergo multi-step oxidative processes to form multifunctional
products, which have sufficiently low volatility to contribute to SOA via gas-particle partitioning (Hallquist
et al., 2009; Atkinson and Arey, 2003).



Research has shown that secondary aerosol (SA) makes a significant contribution (30−77%) to $PM_{2.5}$
during the severe haze events in China (Huang et al., 2014; Guo et al., 2014; Jimenez et al., 2009). However,
there still exists a significant gap between the predicted SA derived from the current atmospheric quality
models and that observed in field observations (Zhao et al., 2018; Yang et al., 2018; Zheng et al., 2015).
Therefore, considering the characteristics of complex pollution in China, it is crucial to study the synergistic
effects of $SO_2$ and $NH_3$ on the formation of SA, which might be an outstanding source of SA formation
(Zhao et al., 2018; Chu et al., 2016; Liu et al., 2016; Santiago et al., 2012; Na et al., 2007).
A few studies have focused on the influence of $SO_2$ or $NH_3$ on SA formation. It has been found that
$SO_2$ promotes SA formation from typical biogenic (e.g., isoprene and α-pinene) and anthropogenic (e.g.,
toluene, o-xylene, 1,3,5-trimethylbenzene, and gasoline vehicle exhaust) precursors through acid-catalyzed
reactions (Chu et al., 2016; Liu et al., 2016; Lin et al., 2013; Santiago et al., 2012; Jaoui et al., 2008;
Kleindienst et al., 2006; Edney et al., 2005), which promote the reactive uptake process of organic species
or enhance the formation of high-molecular-weight compounds (Liggio and Li, 2008; Liggio et al., 2007;
Liggio and Li, 2006). With regard to the role of $NH_3$ in SA formation, knowledge is still limited. Meanwhile,
inconsistent impacts of $NH_3$ on SA formation have been reported under different precursor systems. For
example, $NH_3$ could elevate SA formation in the α-pinene/ozone oxidation system through acid-base
reactions (Na et al., 2007), while the effects of $NH_3$ neutralization were masked by other multiple factors
and did not show significant influence on isoprene-derived SOA formation (Lin et al., 2013), and addition
of $NH_3$ even significantly reduced the SA formation in the styrene/ozone system, which was caused by
nucleophilic attack from the $NH_3$ molecule leading to rapid decomposition of the major aerosol products (Na
et al., 2006). For the photo-oxidation of aromatic VOCs (e.g., toluene, o-/m-/p-xylene), the presence of $NH_3$
could facilitate new particle formation (NPF) and particle growth, subsequently leading to increased SA





81 formation (Li et al., 2018; Liu et al., 2015b).

82 At the present time, the effects of $SO_2$ and $NH_3$ on SA formation have rarely been studied under highly

83 complex pollution conditions (Chu et al., 2016). Meanwhile, vehicular evaporation emissions have been

84 reported to be non-negligible contributors (39.20 %) to ambient VOCs from anthropogenic sources

85 compared with vehicular tailpipe emissions (Liu et al., 2017a). Therefore, it is necessary to study the

86 influence of $SO_2$ and $NH_3$ on SA formation from evaporation emissions.

87 In this study, unburned gasoline vapors were used as a substitute for evaporative emissions, and the

88 roles of $SO_2$ and $NH_3$ on SA formation from the photo-oxidation of unburned gasoline vapors were

89 investigated in a 30 $m^3$ indoor smog chamber, in order to understand the formation potential of SA from

90 oxidation of gasoline vapor in the cocktail of pollutants in Beijing. The respective influences of $SO_2$ and

91 $NH_3$ on both the microphysics and chemistry of SA formation were examined. Meanwhile, the chemical

92 compositions of the formed SOA in the presence of $SO_2$ and $NH_3$ were further explored by applying positive

93 matrix factorization (PMF) analysis. The formation potentials of SA, sulfur- and nitrogen-containing

94 organics from vehicular evaporation emissions in the presence of $SO_2$ and $NH_3$ were evaluated and discussed.

95 ## 2 Materials and Methods

96 ### 2.1 Gasoline fuel

97 The utilized gasoline fuel with grade 92# was collected (refer to the standard Method for manual

98 sampling of petroleum liquids (GB/T 4756-2015)) from a gas station located in Beijing. The gasoline

99 complies with the China V gasoline fuel standard. It contains 22.8 % (v/v) aromatics (mainly including

100 benzene, toluene, xylene, trimethylbenzene) and 12.1 % (v/v) olefins. The composition of the gasoline is

101 similar to the gasoline collected in North China reported by Tang et al. (2015) and could represent the

102 gasoline used in most areas of China for studying SA formation potential. Details of the gasoline composition



are given in Table S1.
**2.2 Smog chamber facility**
A series of photochemical experiments with unburned gasoline vapors in the absence or presence of
$SO_2$ or $NH_3$ were performed in a 30 $m^3$ indoor smog chamber at the Research Center for Eco-Environmental
Sciences, Chinese Academy of Sciences (RCEES-CAS). The detailed schematic structure of the indoor smog
chamber is given in Fig. S1 and described elsewhere (Chen et al., 2018). Briefly, the cuboid chamber reactor
(L × W × H = 3.0 × 2.5 × 4.0 m, S/V = 1.97 $m^{-1}$) was irradiated by 120 UV lamps (Philips) with peak
intensity at 365 nm, providing a $NO_2$ photolysis rate of 0.55 $min^{-1}$. The interior was coated with 125 μm-
thick FEP100 film (DuPont[TM], US) and the chamber was located in a temperature-controlled room, in which
the temperature (T) and relative humidity (RH) could be controlled mechanically. Meanwhile, a three-wing
stainless-steel fan coated with Teflon was installed inside the reactor to guarantee that the gas and particle
phase species mix sufficiently before photochemical reaction.
The chamber was also equipped with a series of gas- and particle-phase monitoring instruments. For
gaseous $NO_x$, $O_3$ and $SO_2$, a chemiluminescence $NO_x$ analyzer (Model 42i-TL, Thermo Fisher Scientific,
USA), a UV photometric $O_3$ analyzer (Model 49i, Thermo Fisher Scientific, USA) and a pulsed fluorescence
$SO_2$ analyzer (Model 43i, Thermo Fisher Scientific, USA) were used to monitor the concentrations in real
time, respectively. The VOC species in gasoline were measured with a gas chromatograph (7890B GC,
Agilent, USA) equipped with a DB-624 column (60 m × 0.25 mm × 1.40 μm, Agilent, USA) and a mass
spectrometry detector (5977A MS, Agilent, USA) (GC-MS). In addition, high-resolution time-of-flight
proton transfer reaction mass spectrometry (HR-ToF-PTRMS) (Ionicon Analytik GmbH, Austria) was also
used for the measurement of gas-phase hydrocarbons and their intermediate products. The size distribution
and number concentration of the formed particulate matter (PM) were measured using a scanning mobility



particle sizer (SMPS, TSI, USA), which was composed of a differential mobility analyzer (DMA, 3080
Classifier, TSI, USA) coupled with a condensation particle counter (CPC, 3776, TSI, USA). The mass
concentration was estimated based on the volume concentration and the density of PM calculated from the
equation $\rho = d_{va}/d_m$, where $d_{va}$ is the mean vacuum aerodynamic diameter measured by an Aerodyne High-
Resolution Time-of-Flight Aerosol Mass Spectrometer (HR-ToF-AMS) and $d_m$ is the mean electrical
mobility diameter measured by SMPS (DeCarlo et al., 2004). The calculated density of PM ranged from 1.5
to 1.6 g cm$^{-3}$ in the different reaction systems. The mass concentration and chemical composition of PM
were simultaneously monitored using a high-resolution time-of-flight aerosol mass spectrometer (HR-ToF-
AMS, Aerodyne Research Inc. USA). Meanwhile, T and RH were monitored real-time using a hydro-
thermometer (Vaisala HMP110) during the entirety of each experiment.
**2.3 Wall loss corrections**

The measured particle concentration was corrected in accordance with the relationship between the

deposition rate ($k_{dep}$) and particle diameter ($D_p$, nm) (i.e., $k_{dep} = 4.15 \times 10^{-7} \times D_p^{1.89} + 1.39 \times D_p^{-0.88}$), which
was described by Takekawa et al. (2003). The wall loss rates of NO$_2$, NO, O$_3$ and VOC species were
determined to be $(1.67 \pm 0.25) \times 10^{-4}$, $(1.32 \pm 0.32) \times 10^{-4}$, $(3.32 \pm 0.21) \times 10^{-4}$ and $(2.20 \pm 0.39) \times 10^{-4}$ min$^{-1}$
$^1$, respectively. Therefore, the wall loss of gas phase species was evaluated to be less than 5% of their
maximum concentration in our study.

Wall losses of semi-volatile organic compounds (SVOCs) and low-volatility organic compounds

(LVOCs) would lead to a substantial underestimation of SA formation (Krechmer et al., 2016; Ye et al., 2016;
Zhang et al., 2015; Zhang et al., 2014), which is caused by the competition between these vapors condensing
onto particles versus onto chamber walls. This competition could be evaluated by the corresponding
timescales associated with reaching gas-to-particle partitioning equilibrium ($\bar{\tau}_{g-p}$) and vapor wall loss ($\tau_{g-w}$)





(Zhang et al., 2014), and this underestimation of SA formation could be approximately quantified by the
ratio of these two timescales (i.e., $\bar{\tau}_{g\text{-}p}/\tau_{g\text{-}w}$). According to the methods described by Zhang et al. (2014), $\bar{\tau}_{g\text{-}p}$
and $\tau_{g\text{-}w}$ could be estimated assuming an upper bound and a lower bound of the molecular mass of organic
vapors (MW) (100−300 g mol$^{-1}$) (as discussed in Supporting Information). In order to accurately quantify
the SA formation, the underestimation caused by the loss of SVOCs and LVOCs (include sulfuric acid gas)
to the chamber walls was taken into account in this study. In our study, the SA yields were underestimated
by a factor of 1.97−2.82 fold when considering the ratio of these two timescales (i.e., $\bar{\tau}_{g\text{-}p}/\tau_{g\text{-}w}$), which
showed a decreasing trend with the increase of the $SO_2$ and $NH_3$ initial concentrations, suggesting that an
increasing proportion of vapors is partitioned onto the suspended particle surface rather than the chamber
wall.
**2.4 Experimental conditions**

Prior to each experiment, the chamber reactor was flushed by purified and dry zero air for about 24−36

h at a flow rate of 100 L min$^{-1}$ until almost no gas-phase species (i.e., $NO_x$, $O_3$ and $SO_2$) could be detected
(< 1 ppb) and the particle number concentration was < 10 cm$^{-3}$. Before the experiments, the chamber was
humidified to ~50 % RH by passing purified zero air through ultra-pure water (18.2 MΩ, Millipore Milli-
Q). After that, a known volume of liquid gasoline (100 µL) was injected into the chamber through a heated
Teflon line system (~100 °C) carried by purified dry zero air to ensure that all were evaporated into the
chamber. Subsequently, $NO_x$, $SO_2$ or/and $NH_3$ were successively injected into the chamber from standard
gas cylinders using mass flow controllers. The initial VOCs/$NO_x$ ratio (ppbC/ppb) was kept constant (Table
1). In order to reduce the adsorption of $NH_3$ in the pipeline, the $NH_3$ flow in a bypass line was balanced for
about 30 min before it was injected into the chamber. The concentrations of $NO_x$ and $SO_2$ were continuously
monitored until they were stable, ensuring that the gaseous species mixed well in the chamber. For the





concentration of $NH_3$, the value was estimated according to the amount of $NH_3$ introduced and the volume
of the reactor chamber. The experiment was then conducted for about 8 h after turning off the fan and turning
on the UV lights. All the experiments were performed at a temperature of $26 \pm 1$ °C and wet conditions (RH
$= 50 \pm 3$ %). The detailed experimental conditions are listed in Table 1. The letters in the abbreviations
represent the reactants introduced into the chamber reactor for each experiment. For example, SGN is an
experiment with the presence of sulfur dioxide (S), gasoline vapor (G), and nitrogen oxides (N). Four
experiments (Exps. SGN1, SGN2, SGN3, and SGN4) were carried out at different $SO_2$ initial concentrations.
AGN is an experiment with the presence of ammonia (A), gasoline vapor (G), and nitrogen oxides (N). Two
experiments (Exps. AGN1 and AGN2) were carried out at different $NH_3$ initial concentrations.
## 3 Results and discussion
### 3.1 Effect of $SO_2$ and $NH_3$ on the gas-phase precursors
Time-resolved concentrations of inorganic and organic gas-phase species during the photo-oxidation of
gasoline/$NO_x$ in the absence or presence of $SO_2$ and $NH_3$ are shown in Fig. S2 and Fig. S3, respectively.
After turning on the UV lights, NO was rapidly converted to $NO_2$. At the same time, $O_3$ was gradually
generated, with a maximum concentration of up to 350 ppb (Fig. S2). As shown in Fig. S2, there was no
obvious difference in the variation of $NO_x$ and $O_3$ in the presence of $SO_2$ or $NH_3$. Meanwhile, the decay of
typical VOC precursors (e.g., benzene, toluene) measured by HR-ToF-PTRMS is given in Fig. S3, which
traced very closely with the GC-MS results (Fig. S4). There were also no observable differences in these
VOCs among these experiments. According to the decay curves of aromatic hydrocarbons, the OH radical
concentrations were estimated to be $(7.54-8.40) \times 10^6$ molecules $cm^{-3}$, which were also similar among these
experiments. In addition, the typical mass spectra of organic gas-phase species derived from HR-ToF-
PTRMS after 480 min of the photo-oxidation reaction at different concentrations of $SO_2$ or $NH_3$ are shown



in Fig. S5, and no significant differences were found. Therefore, it is reasonable to deduce that the presence
of SO₂ or NH₃ did not significantly impact the initial gas-phase oxidation mechanism of gasoline. This was
consistent with the previous study conducted by Chu et al. (2016), who found that the presence of SO₂ and
NH₃ did not significantly impact the initial gas-phase oxidation of toluene in the presence of NOₓ.

**3.2 Role of SO₂ in secondary aerosol formation**

To investigate the effects of SO₂ on SA formation from the photo-oxidation of gasoline/NOₓ, smog

chamber experiments with different SO₂ initial concentrations were carried out (Table 1). As shown in Fig.
1, compared to the experiments without the addition of SO₂, the SA concentration was enhanced to different
degrees (1.6−2.6 times) in the presence of different SO₂ concentrations (35−151 ppb, i.e., 100−431 μg m⁻³).
As for each chemical species (i.e., organics, nitrate, sulfate, and ammonium), they all showed a trend of
linear increase with the increase of SO₂ concentration (Fig. 2), which indicated that aerosol formation will
be significantly promoted by the existence of SO₂, especially for the sulfate and organic aerosol. Previous
studies have also revealed its promoting role on SA formation from different precursors (Zhao et al., 2018;
Liu et al., 2017b; Díaz-de-Mera et al., 2017; Liu et al., 2016; Chu et al., 2016). In addition, it is worth noting
that ammonium aerosols were formed without the addition of gaseous NH₃ (Fig. S6), which indicated that
some NH₃ was present in the background air in the chamber (Liu et al., 2015c). According to the
concentration of generated ammonium aerosols, the concentration of background NH₃ was estimated to be
~15 ppb using the E-AIM model (Clegg and Brimblecombe, 2005; Wexler and Clegg, 2002; Clegg et al.,
1998). Therefore, for the experiments with the presence of NH₃, the concentration of injected NH₃ (150−200
ppb) was much higher than this value to identify the effect of NH₃ on SA formation.

Additionally, the particle number concentrations and size growth were greatly enhanced by the presence

of SO₂. As evident from Fig. 3, the corresponding maximal particle number concentrations (5.82 × 10⁴−1.91



$\times 10^5$ # cm$^{-3}$) were significantly enhanced by a factor of 2.9−3.3 in the presence of SO$_2$. This universal
phenomenon has been reported by many studies (Díaz-de-Mera et al., 2017; Liu et al., 2017b; Liu et al.,
2016; Chu et al., 2016). For example, the maximal particle number concentrations were enhanced by one
order of magnitude in the presence of SO$_2$ (~130 ppb) in the photo-oxidation of high concentration
toluene/NO$_x$ (Chu et al., 2016). For complex precursor systems, Liu et al. (2016) have also found that under
high SO$_2$ concentration (~150 ppb) conditions, the maximum particle number concentrations increased by
5.4−48 times compared to those without SO$_2$ during the photo-oxidation of gasoline vehicle exhaust. In
addition, size distributions of generated SA in smaller size ranges (4−160 nm) were also determined using
another SMPS equipped with a nanometer differential mobility analyzer (Nano-DMA), indicating that the
new particle formation (NPF) phenomenon was enhanced significantly when the SO$_2$ concentration
increased (Fig. S7). The presence of high concentrations of SO$_2$ would generate sulfuric acid (H$_2$SO$_4$), which
would contribute to nucleation and increase the total particle number concentrations (Zhao et al., 2018; Sipilä
et al., 2010). Meanwhile, as the SO$_2$ concentration increased from 35 ppb to 151 ppb, the maximal particle
diameters (144−172 nm) became larger, which will have a direct impact on the scattering and absorption of
light (Seinfeld and Pandis, 2016). An enhancement effect of SO$_2$ on the surface area of particles was also
observed. As shown in Table 1, the surface area of aerosol particles at the end of each experiment increased
from $1.12 \times 10^3$ to $2.46 \times 10^3$ μm$^2$ cm$^{-3}$ when the SO$_2$ concentration increased from 0 to 151 ppb. The larger
surface area would be beneficial to the condensation and heterogeneous uptake of low-volatility vapors
(Chapleski et al., 2016), consequently leading to higher SA yield in the presence of SO$_2$ (Table 1) (Santiago
et al., 2012).

In order to further investigate the role of SO$_2$ in the chemistry of SOA formation, the particle acidities

were estimated using the E-AIM model (Model II: H$^+$ - NH$_4^+$ - SO$_4^{2-}$ - NO$_3^-$ - H$_2$O) (Clegg and Brimblecombe,





2005; Wexler and Clegg, 2002; Clegg et al., 1998). The concentrations of chemical components (i.e., $NH_4^+$,
$SO_4^{2-}$, and $NO_3^-$) at the time when the SOA formation rate reached its peak were used as the inputs of the
model. As shown in Fig. 4, the $H^+$ concentration was increased from 8.5 to 32.5 nmol m$^{-3}$ with the increase
of $SO_2$ concentration under moderate humidity conditions (RH = 50 %) and the higher SOA concentration
and SOA yield could be well explained by the enhancement of the particle acidities ($R^2$ = 0.960 and $R^2$ =
0.986, respectively). This phenomenon was related to the well-known acid-catalyzed reactions of
multifunctional aldehydes (e.g., glyoxal and methylglyoxal), which were the products of aromatic
hydrocarbons in the gasoline vapors through the gas-phase photo-oxidation. Previous studies have reported
that hemiacetals, acetals and alcohols could be generated through the acid-catalyzed heterogeneous reactions
of glyoxal (Czoschke et al., 2003; Jang et al., 2002). These low-vapor-pressure products preferentially
partition into the particle phase and subsequently contribute to the SOA formation (Cao and Jang, 2007;
Casale et al., 2007; Jang et al., 2002).
In addition, the sulfur-containing organics formed in the presence of $SO_2$ might be another reason for
the increase of SOA yield (Kundu et al., 2013; Liggio et al., 2005). According to the linear fitting between
the concentration of formed $SO_4^{2-}$ and the amount of consumed $SO_2$ (after wall loss correction for $SO_2$,
sulfuric acid gas and sulfate), there was a large gap between the slope of the line and the ratio of $M(SO_4^{2-})$
and $M(SO_2)$, as shown in Fig. S8. There are some possible reasons for this, including the underestimation of
deposition and heterogeneous reaction of sulfur species on the wall, the formation of organic sulfur-
containing products, and small leaks of pollutants from the smog chamber. Jaoui et al. (2008) also reported
that the acidic aerosol generated in the presence of $SO_2$ could lead to sulfur-incorporating reactions in the
particle phase during the photo-oxidation of α-pinene/toluene/$NO_x$ mixtures. Sulfur-containing organics
could be generated via reactions of organic species (e.g., polycyclic aromatic hydrocarbons (PAHs),





C10−C12 alkanes, alcohols, epoxides) with sulfate, bisulfate or sulfuric acid, especially under high relative
humidity and acidity conditions (Riva et al., 2015, 2016; Huang et al., 2015; Hatch et al., 2011; Surratt et al.,
2007; Liggio et al., 2005). Huang et al. (2015) have revealed that sulfur-containing organics with R-O-SO$_3^-$
functional groups will yield S-bearing organic fragments ($C_xH_yO_zS$) during ionization, which subsequently
could be detected by HR-ToF-AMS and used as marker ions to quantify them. In our gasoline/NO$_x$
experiments in the presence of SO$_2$, the ions CSO$^+$, CH$_3$SO$_2^+$ and CH$_3$SO$_3^+$ could be separated (Fig. S9),
although uncertainty might be induced in the peak-fitting of the highly abundant ions C$_2$H$_4$O$_2^+$, C$_6$H$_7^+$, and
C$_5$H$_3$O$_2^+$. These characteristic ions (i.e., CSO$^+$, CH$_3$SO$_2^+$ and CH$_3$SO$_3^+$) also have been observed from sulfur-
containing organics in previous field measurements (Huang et al., 2015; Farmer et al., 2010). According to
the estimation method for sulfur-containing organics mentioned in Huang et al. (2015), we found that the
signal of these ions and the concentrations of sulfur-containing organics increased with the SO$_2$ initial
concentration (Fig. 5). The estimated concentrations of sulfur-containing organics (13−26 ng m$^{-3}$) were
comparable to those (~ 20 ng m$^{-3}$) observed in the mid-Atlantic United States, which were derived from
biogenic and anthropogenic hydrocarbons (Meade et al., 2016). Therefore, photo-oxidation of gasoline vapor
in the presence of SO$_2$ might be a noteworthy source of sulfur-containing organics, although the
concentration was very low compared to that of generated SO$_4^{2-}$ (~ 0.1% of SO$_4^{2-}$).
**3.3 Role of NH$_3$ in secondary aerosol formation**
Similarly, the role of NH$_3$ in SA formation was examined. The SA concentration was enhanced by a
factor of 2.0−2.5 in the presence of NH$_3$, as shown in Fig. S10a. The formation of SOA, NO$_3^-$ and NH$_4^+$ was
enhanced to varying degrees. The increase of NO$_3^-$ and NH$_4^+$ could be attributed to the formation of inorganic
NH$_4$NO$_3$ in the presence of NH$_3$. The NO$^+$/NO$_2^+$ ratio, which could be derived from HR-ToF-AMS, has
often been used as a proxy for identification of inorganic nitrate and organic nitrogen compounds (Farmer





et al., 2010; Sato et al., 2010; Rollins et al., 2009). Generally, the $NO^+/NO_2^+$ ratio of inorganic nitrate
(1.08−2.81) is lower than that of organic nitrogen compounds (3.82−5.84) (Liu et al., 2016). In this study,
the $NO^+/NO_2^+$ ratio became substantially lower (~ 2.00) in the presence of $NH_3$ compared with that in the
absence of $NH_3$ (~ 5.46). This phenomenon further indicated that $NH_4NO_3$ became a dominant nitrate species
in the presence of $NH_3$. As for the reason for SOA enhancement, the presence of $NH_3$ could react with some
organic acids and subsequently contribute to SOA formation (Na et al., 2007; Na et al., 2006), which could
be supported by the increase of N/C (from 0.016 to 0.033) with increasing $NH_3$ concentration at similar
concentrations of $NO_x$. This result indicated that $NH_3$ was incorporated in the photo-oxidation of gasoline
vapor. In addition, we have found that the presence of $NH_3$ readily increased the particle diameter and
number concentration of SA generated in the photo-oxidation of gasoline (Figs. S10b and S10c). These
phenomena indicated that $NH_3$ played an important role in new particle formation (NPF). These results are
consistent with the simulation results finding that $NH_3$ promotes atmospheric NPF and also the conversion
of $SO_2$ and $NO_2$ (Jiang and Xia, 2017). Meanwhile, increased surface area of particles was also observed
(Table 1, $2.07 \times 10^3$ and $2.48 \times 10^3$ $\mu m^2$ $cm^{-3}$) as the $NH_3$ concentration increased from 0 to 150 and 200
ppb. Similarly, the larger surface area would favor the partitioning of low-volatility vapors to the particle
phase, leading to the higher SA yield (Table 1).
Previous studies have reported that the reaction of carbonyl compounds (e.g., glyoxal) could be
catalyzed by $NH_4^+$ ions through a Bronsted acid pathway or an iminium pathway, which could generate N-
containing products and oligomers (Nozière et al., 2009). It has been reported that nitrogen-containing
organics could contribute a substantial fraction to SOA (Liu et al., 2015c; Farmer et al., 2010; Cheng et al.,
2006). Previous researchers have identified the characteristic fragments of nitrogen-containing organics as
$C_xH_yN_n$ and $C_xH_yO_zN_n$ using HR-ToF-AMS (Lee et al., 2013; Farmer et al., 2010; Galloway et al., 2009).





In our study, the typical normalized mass spectrum of N-containing fragments in SOA after 480 min of
photo-oxidation reaction at different concentrations of $NH_3$ are given in Fig. 6. The prominent peaks in the
$C_xH_yN_n$ family were at $m/z$ 27 ($CHN^+$), 30 ($CH_4N^+$), 40($C_2H_2N^+$), 41($CHN_2^+$, $C_2H_3N^+$), 42($C_2H_4N^+$),
43($C_2H_5N^+$), 54($C_2H_2N_2^+$, $C_3H_4N^+$), 55($C_3H_5N^+$), and 68($C_3H_4N_2^+$, $C_4H_6N^+$); and the $C_xH_yO_zN_n$ fragments
were dominated by 45($CH_3ON^+$), 46($CH_4ON^+$), 59($C_2H_5ON^+$), 63($CH_5O_2N^+$), 73($C_2H_5ON_2^+$, $C_3H_7ON^+$),
86($C_3H_4O_2N^+$, $C_3H_6ON_2^+$), 91($C_3H_9O_2N^+$), 97($C_4H_5ON_2^+$), and 104($C_3H_6O_3N^+$, $C_4H_{10}O_2N^+$). The N-
containing fragments observed in the experiment without added $NH_3$ could be attributed to the reactions
between organic peroxy ($RO_2$) radicals and $NO_x$ (Arey et al., 2001) or uptake of background $NH_3$ by SOA.
Additionally, it was obvious that the signal intensities of most N-containing fragments became significantly
stronger as the $NH_3$ concentration increased (150−200 ppb). Therefore, a considerable amount of nitrogen-
containing organics (the ratio of nitrogen-containing organics to SOA was about 6.7−7.7%) was formed
during the photo-oxidation of gasoline vapor in the presence of $NH_3$. This was consistent with the previous
study conducted by Liu et al. (2015c), who observed the formation of organic nitrogen compounds in the
SOA generated from the OH oxidation of m-xylene (Liu et al., 2015c). Meanwhile, the promoting role of
$NH_3$ in the formation of N-containing species was also observed in the reaction system of ozonolysis and
photo-oxidation of α-pinene (Babar et al., 2017).
In addition, elemental analysis was also carried out to elucidate the SOA chemical composition and
SOA formation mechanisms (Chhabra et al., 2011; Heald et al., 2010) at different concentrations of $NH_3$.
The time evolution of H/C and O/C in SOA formed from the photo-oxidation of gasoline vapor at different
concentrations of $NH_3$ is shown in Fig. 7. As evident from Fig. 7, all data points are located in the triangular
area for slope between -1 and 0, which suggests that SOA formation from the photo-oxidation of gasoline
vapor is a combination of carboxylic acid and alcohol/peroxide (Heald et al., 2010). Meanwhile, in the


presence of NH$_3$, as shown in Fig. 8, N/C increased as reaction proceeded in the initial oxidation stage
(0−120 min), accompanied by a rapid increase of O/C (0.12−0.67), a decrease of H/C (2.12−1.61), and a
rapid formation of SOA. During this stage, the photo-oxidation of VOC precursors leads to a rapid increase
in O/C and a rapid decrease in H/C. The termination chemistry of NO$_x$ with free radicals and the NH$_3$ uptake
result in a rapid increase in N/C. As the reaction proceeded further (120−300 min), an increase of H/C which
should be caused by NH$_3$ uptake resulted in an almost constant oxidation state of SOA in the continuous
photo-oxidation, accompanied by an increase in the SOA concentration. Nozière et al. (2009) have reported
that N-containing products would be generated from carbonyl compound (e.g., glyoxal) self-reactions
catalyzed by ammonium ions, which will have a dramatic impact on the volatility of oxidation products and
the yield of SOA (Ortiz-Montalvo et al., 2014). In the last stage of the reaction (360−480 min), NH$_3$ uptake
might reach saturation; therefore, H/C and N/C are almost constant. Comparing experiments with different
concentrations of NH$_3$, the average H/C shows an obvious increase (1.53−1.70) while the average O/C
(0.70−0.78) shows a slight increase with the increase of NH$_3$ concentration (0−200 ppb), seen in Fig. S11.
The slope in the Van Krevelen diagram shows a trend from slope = -1 to slope = 0 (Fig. S11), indicating that
the formed carboxylic acid would further react with NH$_3$ via acid-base reaction to generate an ammonium
salt of a carboxylate anion in the presence of NH$_3$ (Na et al., 2007). Meanwhile, Xu et al. (2018) recently
found that imidazole products containing multiple oxygen atoms could be generated through heterogeneous
reactions between NH$_3$ and carbonyl compounds (e.g., glyoxal) (Xu et al., 2018), which might also contribute
to the increase in the O/C of the SOA.
**3.4 Different roles of SO$_2$ and NH$_3$ in SOA chemical properties**

The chemical properties of the SOA generated under the different concentration of SO$_2$ or NH$_3$ were

further compared by applying positive matrix factorization (PMF) analysis to the HR-ToF-AMS data,





respectively (Chu et al., 2016; Liu et al., 2014). The details of PMF analysis are given in the Supporting
Information. For the experiments under different $SO_2$ concentration conditions (i.e., Exps. GN, SGN1, SGN2,
SGN3 and SGN4), two factors (Factor 1-S and Factor 2-S, Fig. S12a) were identified from the PMF analysis,
and the difference mass spectra ($m/z$ 12−170) between the two factors and the time series of the mass
concentrations are shown in Fig. 9. The intensity of $C_xH_y$ and S-bearing organic fragments ($C_xH_yO_zS$) in
Factor 1-S was obviously stronger than that in Factor 2-S. Meanwhile, fragments in the high $m/z$ range (>
110 Da) were more abundant in Factor 1-S (Fig. 9a, marked in red box). By contrast, the fragments
containing oxygen in Factor 2-S were more abundant than in Factor 1-S, such as the typical fragment $CO_2^+$
($m/z$ 44). Therefore, Factor 1-S was tentatively assigned to the less-oxygenated organic aerosol and
oligomers, while Factor 2-S was more-oxygenated organic aerosol (Ulbrich et al., 2009). Similarly, for the
experiments at different $NH_3$ concentration (i.e., Exps. GN, AGN1 and AGN2), two factors (Factor 1-N and
Factor 2-N, Fig. S12b) were also identified in the same way. According to Fig. 10, Factor 1-N was tentatively
assigned to the less-oxygenated organic aerosol and oligomers, while Factor 2-N was more-oxygenated
organic aerosol and nitrogen-containing organics.

As shown in Fig. 9b and Fig. 10b, these two factors both had different time series during the entire

reaction. With respect to Exps. GN, SGN1, SGN2, SGN3 and SGN4, Factor 1-S was formed later (~ 30 min)
than Factor 2-S, and then continuously increased during the entire reaction. Comparing experiments with
different $SO_2$ concentrations, the maximum concentration of Factor 1-S, which was related to the less-
oxygenated organic aerosol and oligomers, was enhanced with increased $SO_2$ concentration ($R^2 = 0.881$, Fig.
9c). This suggested that the presence of $SO_2$ was prone to decrease the oxidation state of organic aerosol via
acid-catalyzed reactions and enhance the formation of oligomers (Liu et al., 2016), which was consistent
with the evolution of O/C vs. H/C shown in Fig. S13. Meanwhile, the gradually increasing concentration of



Factor 1-S was related to the formation of sulfur-containing organics in the presence of $SO_2$ (Blair et al.,
2017). By contrast, Factor 2-S was first gradually increased with the progress of the reaction and then
decreased after reaching a peak (i.e., inflection point). Meanwhile, the time to reach the inflection point was
affected by the $SO_2$ concentration (Fig. 9b). As the initial concentration of $SO_2$ increased from 0 ppb to 151
ppb, the time corresponding to the inflection point decreased, which indicated that the adverse influence of
acid catalysis on Factor 2-S was gradually enhanced. In addition, the maximum concentration of Factor 2-S
was negatively related with $SO_2$ concentration ($R^2 = 0.987$, Fig. 9c); this suggested that the presence of $SO_2$
and acid catalysis was adverse to the formation of more-oxygenated organic aerosol, leading to the decrease
of the oxidation state of organic aerosol (Fig. S13).

376   By contrast, for Exps. GN, AGN1 and AGN2, Factor 1-N was first increased with the progress of the

reaction and then gradually decreased after reaching a peak (Fig. 10b); while Factor 2-N was formed later
(~ 30 min) than Factor 1-N, and then continuously increased during the entire reaction. This phenomenon
was consistent with the expected behavior, that less-oxidized organic aerosol would be further oxidized to
form more-oxidized organic aerosol. When comparing experiments with different $NH_3$ concentrations, it
was observed that the concentration of Factor 1-N increased with increasing $NH_3$ concentration. Meanwhile,
Factor 1-N, which was related to the more-oxidized organic aerosol and nitrogen-containing organics, was
a dominant factor in the presence of $NH_3$, and its maximum concentration was enhanced with the increase
in $NH_3$ concentration ($R^2 = 0.988$, Fig. 10c). This phenomenon indicated that the formation of more-
oxygenated organic aerosol and nitrogen-containing organics was enhanced with the increase of $NH_3$
concentration. Meanwhile, a negative correlation was observed between the maximum concentration of
Factor 1-N and $NH_3$ concentration ($R^2 = 0.876$, Fig. 10c); this revealed that less-oxygenated organic aerosol
was gradually transformed to highly oxidized species and nitrogen-containing organics in the presence of



$NH_3$.

## 4 Conclusions

In our study, SA formation from the photo-oxidation of gasoline/$NO_x$ in the presence of $SO_2$ or $NH_3$
was investigated. Our experimental results suggested that SA was enhanced by a factor of 1.6−2.6 or 2.0−2.5,
respectively, with the increase of $SO_2$ or $NH_3$ concentration (0−151 ppb and 0−200 ppb, respectively).
Meanwhile, both secondary organic aerosol (SOA) and secondary inorganic aerosol (SIA) could be increased
by varying degrees. In the presence of $SO_2$, $SO_4^{2-}$ showed the most sensitive linear increase trend with the
increase of $SO_2$ concentration (k = $8.4\times10^{-2}$), and SOA was also greatly enhanced (k = $2.9\times10^{-2}$) by the well-
known acid catalytic effect and the formation of sulfur-containing organics. In the presence of $NH_3$, $NH_4NO_3$
was most enhanced, following by organic aerosol. The formation of nitrogen-containing organics, which
were identified by applying PMF analysis, was also promoted by the presence of $NH_3$. Meanwhile,
conspicuous new particle formation (NPF) and particle size growth were observed in the presence of $SO_2$ or
$NH_3$.
Our results indicate that the photo-oxidation of gasoline/$NO_x$ in the presence of $SO_2$ and $NH_3$ is a
significant source of SA. Therefore, in order to mitigate $PM_{2.5}$ pollution in China, emission control strategies
should not only pay attention to primary particulate emissions, but also focus on synergistic reduction of the
emission of SA precursors including $NO_x$, $SO_2$, $NH_3$ and, particularly, VOCs. In this study, a linear
relationship was observed between the SA yield and $SO_2$ or $NH_3$ concentration (Fig. S14). In recent years,
vehicular evaporation emissions have gradually attracted attention due to their non-negligible contribution
(39.20 %, 1.65 Tg $yr^{-1}$) to ambient VOCs (Liu et al., 2017a). Considering the typical concentrations of $SO_2$
and $NH_3$ of 40 ppb and 23 ppb in haze pollution in the north China plain (Cheng et al., 2016), the SA yield
is roughly estimated to be about 0.3. Then, the SA formed from the photo-oxidation of VOCs emitted by



vehicular evaporation in the presence of $SO_2$ and $NH_3$ is roughly estimated to be 0.49 Tg yr$^{-1}$, which is about
twice as much as the primary $PM_{2.5}$ emissions from transportation (0.21 Tg in 2007) in China (Jing et al.,
2015; Zhang et al., 2007). This estimate indicates that vehicular evaporation emissions will be a significant
source of SA in the presence of $SO_2$ and $NH_3$, although the estimate might have a high uncertainty due to
the fact that SA yield might vary considerably under different atmospheric conditions. Meanwhile, in the
presence of $NO_x$, $SO_2$ and $NH_3$, vehicular evaporation emissions might be a potential source of sulfur- and
nitrogen-containing organics, according to the results obtained from our study. Previous studies have
indicated that sulfur- and nitrogen-containing organics have an adverse influence on the climate by light
absorption and/or by affecting aerosol hygroscopicity (Staudt et al., 2014; Nguyen et al., 2012), and they
also have a significant contribution to SOA and nitrogen or sulfur budgets (Lee et al., 2016; Shang et al.,

2016).

Therefore, more attention should be paid to collaborative control reductions in vehicular evaporation
emissions and gaseous pollutants, including $NO_x$, $SO_2$, and $NH_3$. This will contribute to reducing the burden
of $PM_{2.5}$, and then cut the environmental, economic and health costs caused by PM pollution. Corresponding
emission controls should be taken into consideration by policy makers for future management. Our work
will provide a scientific basis for taking corresponding control measures to relieve haze events in China.
Additionally, further work should be focused on SA formation from vehicular evaporation under coexisting
$SO_2$ and $NH_3$ conditions to shed light on the formation mechanism of SA under more atmospherically
relevant conditions.
**Author contributions**
HH, QXM, YCL, and TZC proposed the initial idea. YCL and TZC designed and led the study. YCL,
BWC, QXM, PZ, and TZC conducted the data analyses. TZC, YCL, BWC, PZ, CGL, and JL interpreted the



data. TZC, YCL, JL, and QXM wrote the manuscript, with inputs from all coauthors.

## Acknowledgements

This work was financially supported by the National Key R&D Program of China (2016YFC0202700, 2018YFC0506901), the National Natural Science Foundation of China (41877306, 41877304, 21876185, and 91744205), the special fund of the State Key Joint Laboratory of Environment Simulation and Pollution Control (17L01ESPC), the Youth Innovation Promotion Association, CAS (2018060, 2018055, and 2017064) and Key Research Program of Frontier Sciences, CAS (QYZDB-SSW-DQC018).

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





Table 1. Summary of experimental conditions in this study.

| Exp. [a] | RH (%) | T (°C) | $SO_2$ (ppb) | $NH_3$ [b] (ppb) | $HC_0$ (ppb) | $NO_{x,0}$ (ppb) | $HC_0/NO_{x,0}$ (ppbC ppb$^{-1}$) | Surface [c] ($\mu m^2$ cm$^{-3}$) | $\Delta HC$ ($\mu g$ m$^{-3}$) | $\Delta M$ ($\mu g$ m$^{-3}$) | SA yield [d] |
|---|---|---|---|---|---|---|---|---|---|---|---|
| GN | 50±3 | 26±1 | − | − | 411.0 | 128.4 | 20.61 | $1.12\times10^3$ | 747.8 | 34.6 | 0.130 |
| SGN1 | 50±3 | 26±1 | 35 | − | 419.8 | 121.0 | 22.34 | $1.73\times10^3$ | 871.6 | 58.0 | 0.155 |
| SGN2 | 50±3 | 26±1 | 74 | − | 412.0 | 121.3 | 21.88 | $2.06\times10^3$ | 866.2 | 77.8 | 0.193 |
| SGN3 | 50±3 | 26±1 | 116 | − | 383.6 | 119.8 | 20.62 | $2.23\times10^3$ | 791.1 | 87.1 | 0.226 |
| SGN4 | 50±3 | 26±1 | 151 | − | 394.4 | 125.9 | 20.17 | $2.46\times10^3$ | 810.7 | 106.3 | 0.258 |
| AGN1 | 50±3 | 26±1 | − | 150 | 413.8 | 120.4 | 22.12 | $1.79\times10^3$ | 700.6 | 47.6 | 0.158 |
| AGN2 | 50±3 | 26±1 | − | 200 | 411.5 | 122.6 | 21.61 | $2.23\times10^3$ | 749.1 | 58.3 | 0.166 |

[a] Letters in abbreviations represent the reactants introduced into the chamber reactor, i.e., "G" represents
gasoline, "N" represents nitrogen oxides, "S" represents sulfur dioxide, "A" represents ammonia.
[b] The concentration of $NH_3$ is estimated by the amount of $NH_3$ added and the volume of the smog chamber.
[c] The surface area of aerosol particles measured by SMPS after 480 min of each experiment.
[d] SA yield was calculated after taking vapor and particle wall loss into account.




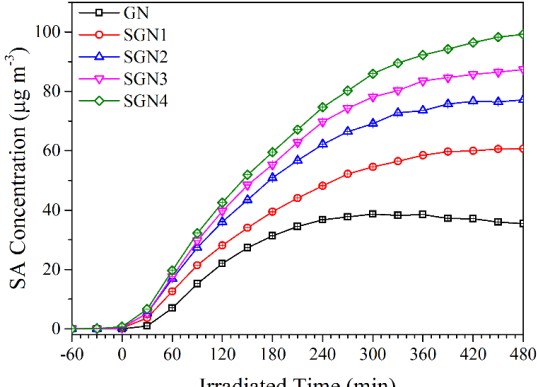


Fig. 1. Time series of secondary aerosol concentrations during the photo-oxidation experiments with different SO$_2$
concentrations (Exps. GN, SGN1, SGN2, SGN3, and SGN4).

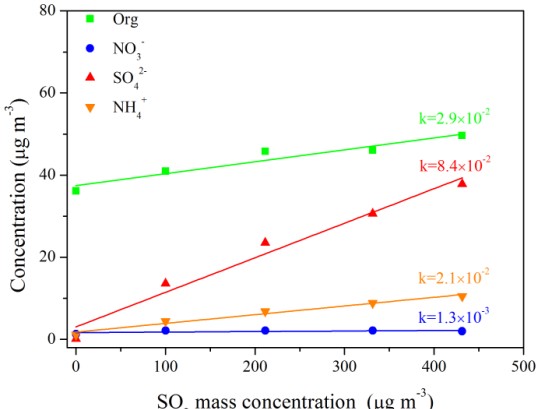


Fig. 2. Linear relationship between the concentration of chemical species (i.e., organic (green), nitrate (blue), sulfate (red), and
ammonium (orange)) and SO$_2$ under different SO$_2$ initial concentration conditions (Exps. GN, SGN1, SGN2, SGN3, and
SGN4). Each line represents a linear fitting and the k values are the corresponding slopes for each chemical species.





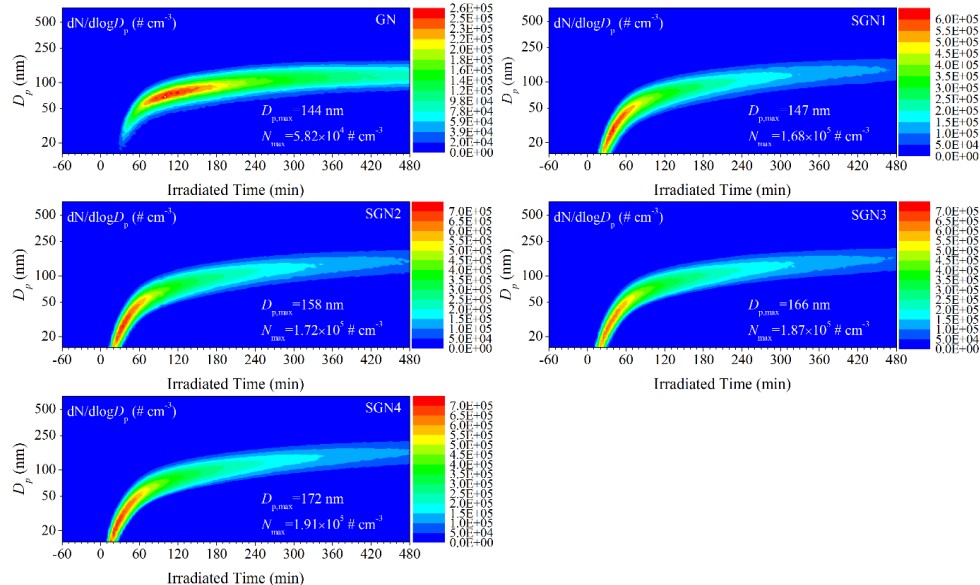


Fig. 3. Time series of the size distributions for the generated secondary aerosol during the photo-oxidation experiments with

different SO$_2$ initial concentrations (Exps. GN, SGN1, SGN2, SGN3, and SGN4). $D_{p,max}$ and $N_{max}$ represent the maximal

diameter and number concentration of generated secondary aerosol, respectively, during each photo-oxidation experiment.

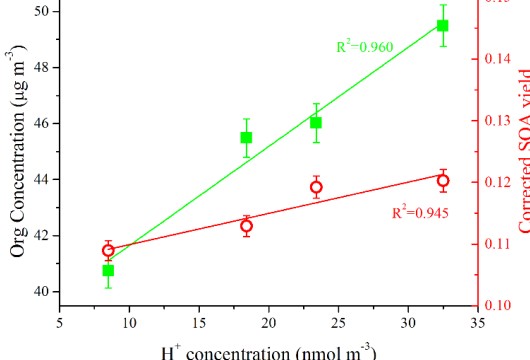


Fig. 4. Relationship between SOA concentration (left y axis), corrected SOA yield (right y axis) and H$^+$ concentration, which

was used to characterize the particle acidities. The H$^+$ concentration presented in this plot was the value when the SOA

formation rate reached the peak during each experiment (Exps. SGN1, SGN2, SGN3, and SGN4).



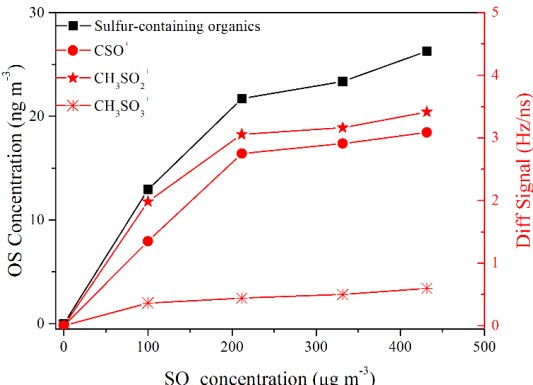


Fig. 5. Signal of fitted peaks, i.e., CSO+, CH₃SO₂+, CH₃SO₃+ (right y axis) and sulfur-containing organics concentration (left
y axis) as a function of SO₂ initial concentration.

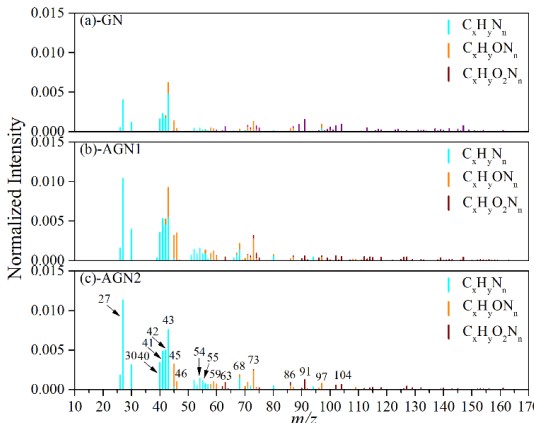


Fig. 6. Typical normalized mass spectra of N-containing fragments in SOA formed from the photo-oxidation of gasoline vapor
at different concentrations of NH₃ (Exps. GN, AGN1 and AGN2).

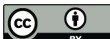



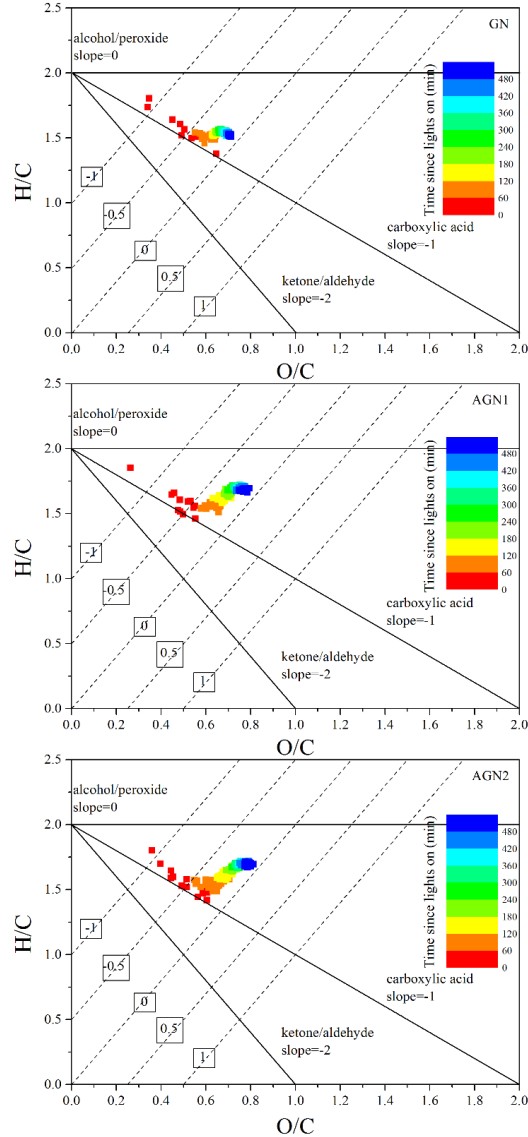


Fig. 7. Time evolution of H/C and O/C in SOA formed from the photo-oxidation of gasoline vapor at different concentrations
of NH$_3$ (Exp. GN, AGN1 and AGN2). The numbers (i.e., -1, -0.5, 0, 0.5, and 1) labeling the dashed lines show the average
carbon oxidation state (OSc $=$ 2×O/C−H/C) (Kroll et al., 2011). The black lines represent the addition of functional groups
to an aliphatic carbon (Heald et al., 2010).





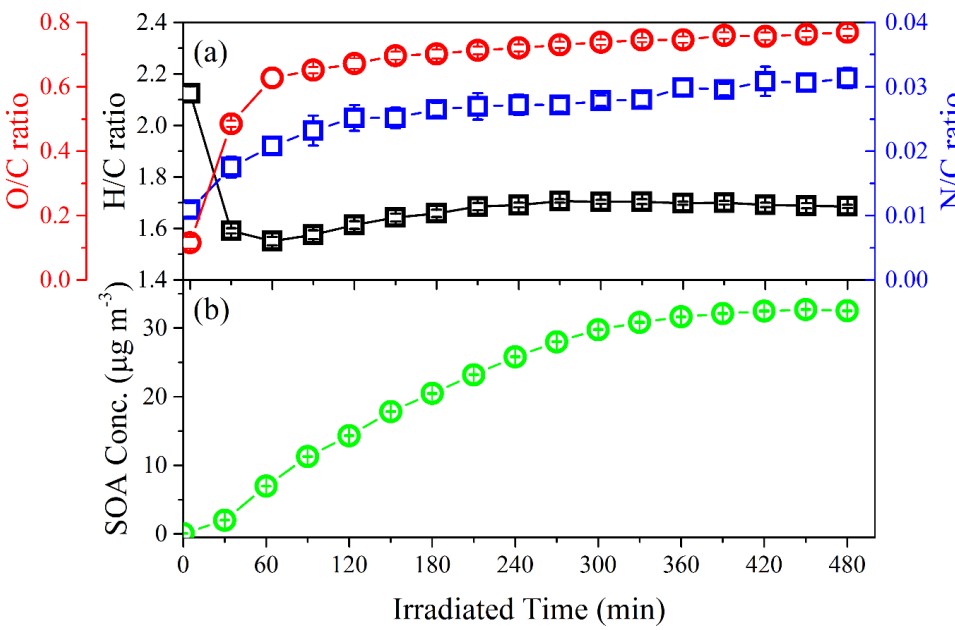


Fig. 8. Time evolution of (a) O/C, H/C and N/C and (b) SOA concentration in the photo-oxidation of gasoline vapor in the
presence of 150 ppb NH$_3$ (Exp. AGN1).





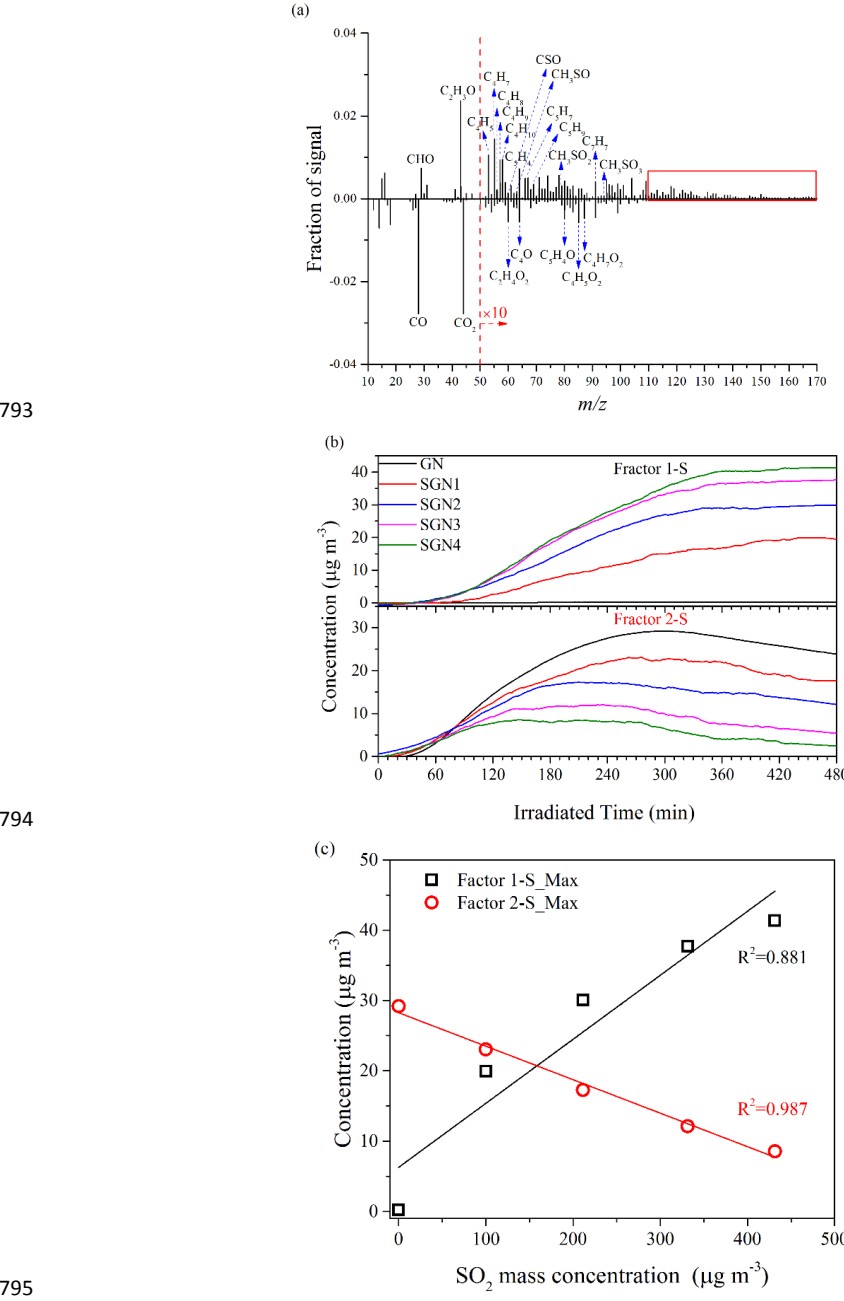




Fig. 9. (a) Difference mass spectra (Factor 1-S−Factor 2-S) between the two factors, (b) Time series of the mass concentration,
and (c) Relationship between the concentration of $SO_2$ and the maximum concentration of the two factors identified by
applying PMF analysis to the AMS data derived from the experiments at different concentrations of $SO_2$ (Exps. GN, SGN1,



SGN2, SGN3 and SGN4).



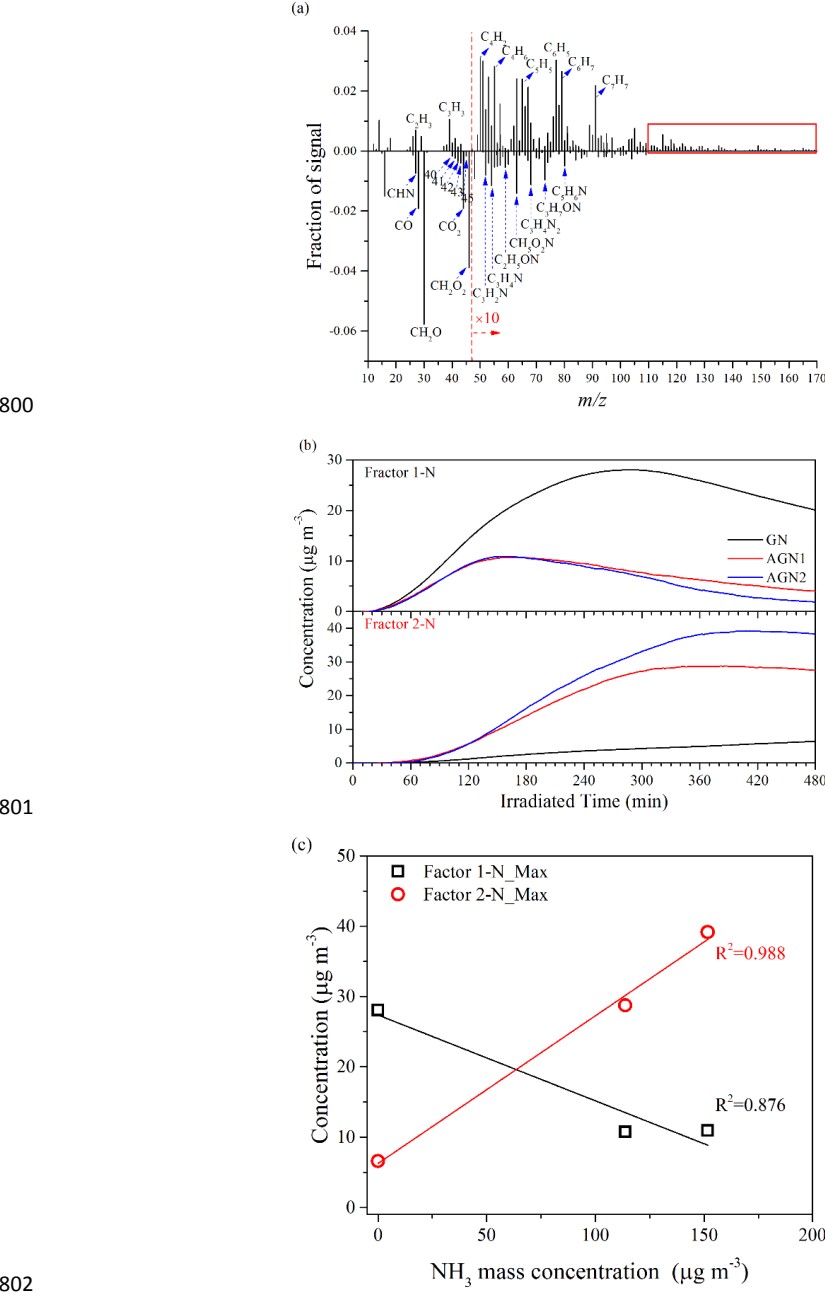




Fig. 10. (a) Difference mass spectra (Factor 1-N−Factor 2-N) between the two factors, (b) Time series of the mass concentration,
and (c) Relationship between the concentration of NH₃ and the maximum concentration of the two factors identified by
applying PMF analysis to the AMS data derived from the experiments at different concentrations of NH₃ (Exps. GN, AGN1



and AGN2).