# Peer review of "Significant source of secondary aerosol: formation from gasoline"

_Atmospheric Chemistry and Physics, 2019_

## Referee Comment (RC1) · Anonymous Referee #1 · 6 Mar 2019

Writing with an easy and fluent style. No obvious grammar problems. Easy to read and understand. However, use some conjunctions and sentence patterns again and again. For example, meanwhile, in addition, this phenomena indicated that... Conclusion is too weak. In other word, do not just use references to explain the normal results. Show the highlight of your own research.

Please also note the supplement to this comment: https://www.atmos-chem-phys-discuss.net/acp-2019-14/acp-2019-14-RC1-supplement.pdf

[Figure]

[Figure]

**Supplement:**

1. Line 21-22: It is good to use specific number to show the results.
2. Line 116-119: "For gaseous NOx, $O_3$ and $SO_2$, a chemiluminescence…" Each gaseous species has been described in detail later, so it is repeated to show NOx, $O_3$ and $SO_2$ here.
3. Line 128: Why use this equation $\rho = d_{va}/d_m$ to calculate mass concentration?
4. Line 133-134: Better to describe RH control system here.
5. Line 164: Pls explain why inject $NO_X$.
6. Line 165: How to combine experimental gasoline vapor with real case?
7. Line 181: Can not find Fig. S2 and S3.
8. Line 186: Can not find Fig. S4.
9. Line 205: Can not find Fig. S6.
10. Line 204-210: ammonium aerosols should be discussed in 3.3.
11. Line 213: Delete #
12. Line 212-213: There were many studies indicated the relationship between $SO_2$ and secondary aerosol. What is the highlight of your experiment?
13. Line 215: Pls revise "were enhanced by one order of magnitude" to "enhanced by XXX to one order of magnitude".
14. Line 240: "This phenomenon" means "the H+ concentration was increased" and
15. Line 247-251: Why use large gap between $SO_4^2$ and $SO_2$ rather than S-bearing organic fragments (CxHyOzS) to show the reason for existing of organic sulfur?
16. Line 282: Too many "This phenomenon indicated".
17. Line 286: Too many "This result indicated that".
18. Line 402-403: Repeated conclusion.
19. Line 422-423: Repeated conclusion.
20. Line 410-412: SA formation still be a small part from vehicular evaporation emissions. What is the meaning of your study?

---

## Referee Comment (RC2) · Anonymous Referee #2 · 26 Mar 2019

Chen et al. propose a study presenting the impact of NH3 and SO2 on secondary organic aerosol formation produced from the oxidation of unburned gasoline vapors. While this study presents interesting results, some aspects (see comments below) remain unclear and/or should be better discussed. One of my main comment is the relevance of the mix of VOC studied in this work when the authors claimed to study relevant unburned gasoline vapors. Indeed, the emission inventory proposed by Liu et al appears very different than the mix of VOC employed in this work. Overall this can lead to an overestimation of the importance of this potential SOA source.

Lines 49-53: If the highest concentrations of NH3 were observed in urban areas, is the

main source from agricultur or vehicle emissions?

Lines 66-70: Split the references and cite proper references. For instance, the work from Kamens and al. should be better acknowledged.

Lines 83-86: The authors should better discuss this section and not overestimate the potential of VTE in producing aerosols. According to Liu et al., 2017, the VOC emitted from the evaporation are mainly dominated by short chain alkane (e.g., Pentane, butane,...) which have a very limited impact on SOA formation. However, the authors mainly discussed/focused on the oxidation of aromatics. In addition, important papers are missing and should be discussed; e.g., Pye and Pouliot 2012, Tkacik et al., 2012.

Lines 121-122: Please change to the widely used acronym (PTR-TOF), (i.e., Yuan et al., 2017 Chem Review).

Lines 130-131: Is the density comparable to other studies?

Lines 191-192: The authors should be careful here. The PTR can measure a certain (small) subset of OVOC produced from a given reaction. Therefore, it is not because the analytical technique was not able to identify any significant differences that mean $SO_2/NH_3$ did not significantly impact the gaseous phase. As an example, a large formation of $H_2SO_4$ is mentioned in the paper. In addition, the authors should propose a more quantitative comparison (as realized with the AMS data) and not only briefly compare the mass spectra.

Line 206: The background of ammonia appears quite subsequent. Do the authors have an explanation? Is it an expected background in chamber experiments?

Fig.3 the color scale is hard to read please modify it.

Lines 213-219: In this section, there are some inconsistencies with the literature. The number (i.e., up to 50 times more particles) reported in earlier studies are different than in the work proposed by Chen et al. However the concentration of $SO_2$ and the precursors were similar. For example, while the $SO_2$ concentration was multiplied by 4

the formation of particles didn't increase significantly. The authors should explain such discrepancy and discuss/compare their results to the existing literature.

Lines 227-232: This is in line with my previous comments. How do they explain such a moderate increase between SGN 1 and 4?

Lines 245-246: Which compounds are the authors referring to? Products from heterogeneous reactions or Aldehydes? In both cases, the sentence should be revised. (i) products formed in the particle phase cannot partition as they are already in the condensed phase and second the aldehydes are not considered as low vapor pressure compounds (Kroll and Seinfeld, 2008).

Lines 249-253: Before speculating on the potential formation of organosulfur the authors should present/discuss the validation of the AMS results: calibration of the AMS, uncertainties of the measurements, SMPS vs AMS data,... This is important in order to claim a large formation of organosulfur.

Lines 265-270: In other words, the large gap between SO2 and SO4 cannot be explained by the organosulfur. The authors should not speculate something and a few lines after conclude that is not the case. In Fig S8, the difference should be tens of ug m-3 while the authors estimated the formation organosulfur of a few tens of ng m-3.

Lines 387-388: The O:C ratio didn't change significantly so I do not think the authors can claim to the formation of highly oxidized species.

---

## Author Comment (AC1) · 27 Apr 2019

**Responses to Referee #1's comments**

**General comment:** Writing with an easy and fluent style. No obvious grammar problems. Easy to read and understand. However, use some conjunctions and sentence patterns again and again. For example, meanwhile, in addition, this phenomena indicated that... Conclusion is too weak. In other word, do not just use references to explain the normal results. Show the highlight of your own research.

**Response:** Many thanks for your constructive comments and valuable suggestions. The manuscript has been carefully revised according to your suggestion both in conjunctions and conclusions. We also added lots of discussion about our experimental results. And the conclusion has been rewritten more concrete in the revised manuscript.

**Revision in the manuscript:**

**Line 28, 89, 123, 154, 260, 344, 367, 391, Delete** "Meanwhile"

**Line 79, Change** "Meanwhile" to "In previous studies"

**Line 205, Change** "Meanwhile" to "Additionally"

**Line 373, Change** "Meanwhile" to "Moreover"

[revised manuscript text omitted]

**Comment 1:** Line 21-22: It is good to use specific number to show the results.

**Response:** Thank you very much for your suggestions. The concentration ranges of $SO_2$ and $NH_3$ have been supplemented in the revised manuscript.

**Revision in the manuscript:**

**Lines 21-23, Change** "It was found that increase in $SO_2$ and $NH_3$ concentrations could promote linearly the formation of SA, which could be enhanced by a factor of 1.6−2.6 and 2.0−2.5, respectively." **to:** "It was found that increase in $SO_2$ and $NH_3$ concentrations (0−151 ppb and 0−200 ppb, respectively) could promote linearly the formation of SA, which could be enhanced by a factor of 1.6−2.6 and 2.0−2.5, respectively."

**Comment 2:** Line 116-119: "For gaseous $NO_x$, $O_3$ and $SO_2$, a chemiluminescence…"

Each gaseous species has been described in detail later, so it is repeated to show $NO_x$,

O$_3$ and SO$_2$ here.

**Response:** Thank you very much. According to your valuable comment, the repeated gaseous species have been deleted in the revised manuscript.

**Revision in the manuscript:**

**Lines 126-128, Delete** the repeated "NO$_x$, O$_3$ and SO$_2$"

**Comment 3:** Line 128: Why use this equation ρ = d$_{va}$/d$_m$ to calculate mass concentration?

**Response:** Thanks for your comment. The equation $\rho = d_{va}/d_m$ mentioned by DeCarlo et al. (2004) was used to calculate the density of PM, and then a relatively reliable PM mass concentration could be calculated based on the measured mobility diameter of particles with the SMPS. Many previous studies have demonstrated that PM effective density could be estimated by comparing mobility diameter from the DMA (i.e., $d_m$) and vacuum aerodynamic diameter from an Aerodyne AMS (i.e., $d_{va}$) in parallel (Jimenez et al., 2003a, b; Bahreini et al., 2005; Alfarra et al., 2006; Ng et al., 2007; Sato et al., 2010). According to this equation, the density of PM was calculated to be 1.5−1.6 g cm$^{-3}$, which was in the range of density of SOA derived from aromatic hydrocarbons (1.24−1.48 g cm$^{-3}$) (Sato et al., 2010) and ammonium nitrate (NH$_4$NO$_3$, 1.72 g cm$^{-3}$) (Bahreini et al., 2005) and could be comparable with the previous studies (Li et al., 2018).

**Comment 4:** Line 133-134: Better to describe RH control system here.

**Response:** Thanks for your comment. The RH control system is achieved by vaporizing Milli-Q ultrapure water contained in a 5.0 L high pressure resistant container and the water vapor is flushed with purified dry zero air into the smog chamber. And the humidification process does not introduce detectable hydrocarbons or particles into the chamber. RH inside the smog chamber could be adjusted from < 5 to 80 %. In this study, ~50 % RH was adjusted prior to each experiment.

**Revision in the manuscript:**

**Lines 153-155, Add:** "As for the RH control system, it is achieved by vaporizing Milli-Q ultrapure water contained in a 5.0 L high pressure resistant container and the water vapor is flushed with purified dry zero air into the chamber."

**Comment 5:** Line 164: Pls explain why inject $NO_x$.

**Response:** Thank you very much. In the experiments, NO was injected into the chamber through the standard gas cylinder (1020 ppm in $N_2$, Beijing Huayuan). Considering there was a small fraction of $NO_2$ (< 1 %) in the standard gas cylinder of NO, so $NO_x$ was used as the proxy for injected NO. In addition, high concentrations of $NO_x$ have been observed in China haze pollution episodes (He et al., 2014; Zou et al., 2015). In order to avoid this misunderstanding, $NO_x$ in the revised manuscript has been replaced by NO.

**Revision in the manuscript:**

**Line 185, Change** "$NO_x$" **to:** "NO"

**Line 188, Change** "$NO_x$" **to:** "NO"

**Comment 6:** Line 165: How to combine experimental gasoline vapor with real case?

**Response:** Thanks for you value comment. In this study, gasoline vapors were used as a substitute for evaporative emissions. In recent years, vehicle ownership in China has increased rapidly, and vehicle-related pollutants (e.g. VOCs) have exacerbated the severity of compound atmospheric pollution, and caused frequent regional haze events in China. Meanwhile, with the implementation of tailpipe exhaust emission control measures, the proportionate share of vehicular evaporative emissions to this pollution has grown, so that it has become a non-negligible contributor (39.20 %) to ambient VOCs from anthropogenic sources (Liu et al., 2017). In addition, the long-chain alkanes (> C6, IVOCs) contained in evaporative emissions would also contribute to SOA formation (Pye and Pouliot, 2012; Tkacik et al., 2012; Presto et al., 2009; Lim and Ziemann, 2005; Zhao et al., 2016). Therefore, the study of the contribution of VOCs emitted by gasoline evaporation to secondary aerosol formation has an important environmental significance for understanding the causes of haze events.

We agree that there might be some differences between the VOCs composition of gasoline vapors directly injected to the smog chamber and evaporative emissions. Thus, further work should be focused on SA formation directly from vehicular evaporative emissions to shed light on the formation mechanism of SA under more atmospherically relevant conditions. Corresponding discussions have been supplemented in the revised manuscript.

**Revision in the manuscript:**

**Lines 476-480, Add:** "Additionally, there might be some differences between the VOCs composition of gasoline vapors directly injected to the smog chamber and vehicular evaporative emissions. Thus, further work should be focused on SA formation directly from vehicular evaporative emissions under coexisting $SO_2$ and $NH_3$ conditions to shed light on the formation mechanism of SA under more atmospherically relevant conditions."

**Comment 7:** Line 181: Can not find Fig. S2 and S3.

**Response:** Thanks for your careful check. Fig. S2 and Fig. S3 could find in the Supplement and corresponding annotation has been added in the revised manuscript.

**Revision in the manuscript:**

**Line 202, Add:** "in the Supplement"

**Comment 8:** Line 186: Can not find Fig. S4.

**Response:** Thanks for your comment. Fig. S4 is shown in the Supplement and corresponding annotation has been added in the revised manuscript.

**Revision in the manuscript:**

**Line 208, Add:** ", in the Supplement"

**Comment 9:** Line 205: Can not find Fig. S6.

**Response:** Thanks for your careful check. Fig. S6 is given in the Supplement and corresponding annotation has been added in the revised manuscript. According to your

Comment 10, Fig. S6 has been changed to Fig. S8 in the revised Supplement.

**Revision in the manuscript:**

**Line 321, Add:** ", in the Supplement"

**Comment 10:** Line 204-210: ammonium aerosols should be discussed in 3.3.

**Response:** Thanks for your suggestion. The corresponding discussion has been moved to Section 3.3 in the revised manuscript.

**Revision in the manuscript:**

**Lines 320-328, Add:** "It is worth noting that ammonium aerosols were formed without the addition of gaseous $NH_3$ (Fig. S8, in the Supplement), which signified that some $NH_3$ was present in the background air in the chamber or introduced during the humidification process of the chamber (Liu et al., 2015c). Unfortunately, appropriate instruments are unavailable to measure the exact concentration of background $NH_3$ in the chamber. According to the concentration of generated ammonium aerosols, the concentration of background $NH_3$ was estimated to be ~15 ppb using the E-AIM model (Clegg and Brimblecombe, 2005; Wexler and Clegg, 2002; Clegg et al., 1998). Therefore, for the experiments with the presence of $NH_3$, the concentration of injected $NH_3$ (150−200 ppb) was much higher than this value to identify the effect of $NH_3$ on SA formation."

**Comment 11:** Line 213: Delete #.

**Response:** Thank you. # has been deleted in the revised manuscript.

**Revision in the manuscript:**

**Line 237, Delete:** "#"

**Comment 12:** Line 212-213: There were many studies indicated the relationship between $SO_2$ and secondary aerosol. What is the highlight of your experiment?

**Response:** Thanks for your comment. You are right. Indeed, there were many previous studies have demonstrated that secondary aerosol (SA) formed from typical biogenic (e.g., isoprene and α-pinene) (Lin et al., 2013; Jaoui et al., 2008; Kleindienst et al., 2006; Edney et al., 2005) and anthropogenic (e.g., toluene, o-xylene, and 1,3,5-trimethylbenzene) precursors (Chu et al., 2016; Liu et al., 2016; Santiago et al., 2012) could be greatly enhanced by the presence of $SO_2$. While, at the present time, the effects of $SO_2$ on SA formation have rarely been studied under highly complex precursors, such as VOCs from vehicular evaporative emissions, especially in China. Previous studied have reported that vehicular evaporative emissions have become non-negligible contributors (39.20 %) to ambient VOCs from anthropogenic sources compared with vehicular tailpipe emissions (Liu et al., 2017). In addition, China has the highest concentration of $SO_2$ in the world due to a large proportion of energy supply from coal combustion (Bauduin et al., 2016). Therefore, in this study, the influence of $SO_2$ on SA formation from evaporative emissions was investigated to simulate the case in the ambient air, which has practical significance for shedding light on the formation mechanism of SA under more atmospherically relevant conditions, especially in China. Meanwhile, the different roles of $SO_2$ and $NH_3$ in secondary organic aerosol (SOA)

chemical properties were further compared by applying positive matrix factorization (PMF) analysis to the HR-ToF-AMS data. Our study indicates that the photo-oxidation of VOCs from vehicular evaporative emissions will be a significant source of SA in the presence of high concentrations of $SO_2$ and $NH_3$. Moreover, these emissions might also be a potential source of sulfur- and nitrogen-containing organics. Our work provides a scientific basis for the synergistic emission reduction of secondary aerosol precursors, including $NO_x$, $SO_2$, $NH_3$ and particularly VOCs, to mitigate PM pollution in China.

**Comment 13:** Line 215: Pls revise "were enhanced by one order of magnitude" to "enhanced by XXX to one order of magnitude".

**Response:** Thanks for your suggestion. This sentence has been revised in the revised manuscript.

**Revision in the manuscript:**

**Lines 239-241, Change** "were enhanced by one order of magnitude in the presence of $SO_2$ (~130 ppb) in the photo-oxidation of high concentration toluene/$NO_x$" **to:** "were enhanced by the presence of $SO_2$ (~130 ppb) to one order of magnitude in the photo-oxidation of high concentration toluene/$NO_x$"

**Comment 14:** Line 240: "This phenomenon" means "the $H^+$ concentration was increased" and.

**Response:** Thanks for your comment. "This phenomenon" means that "the higher SOA concentration and SOA yield could be well explained by the enhancement of the particle

acidities". In order to avoid this misunderstanding, "This phenomenon" was specifically pointed out in the revised manuscript.

**Revision in the manuscript:**

**Line 285, Change** "This phenomenon" **to:** "The higher SOA concentration and SOA yield"

**Comment 15:** Line 247-251: Why use large gap between $SO_4^{2-}$ and $SO_2$ rather than S-bearing organic fragments ($C_xH_yO_zS$) to show the reason for existing of organic sulfur?

**Response:** Thanks very much for your comment. First, we found that there was a large gap between the concentration of formed $SO_4^{2-}$ and the amount of consumed $SO_2$ (after wall loss correction for $SO_2$, sulfuric acid gas and sulfate). As for this large gap, there are might be many possible reasons including the underestimation of deposition and heterogeneous reaction of sulfur species on the wall and the formation of organic sulfur-containing products and so on. In order to confirm the existing of sulfur-containing organics, we further utilized the characteristic ions $CSO^+$, $CH_3SO_2^+$ and $CH_3SO_3^+$ detected by HR-ToF-AMS to quantify the amount of sulfur-containing organics. According to the estimation method for sulfur-containing organics mentioned in Huang et al. (2015), we found that the signal of these characteristic ions and the concentrations of sulfur-containing organics indeed increased with the $SO_2$ initial concentration. Evidences of both two aspects were used to prove the photo-oxidation of gasoline vapor in the presence of $SO_2$ was a noteworthy source of sulfur-containing organics.

In order to more directly explain the presence of sulfur-containing organics, we refer to

your comment, the S-bearing organic fragments ($C_xH_yO_zS$) detected by HR-ToF-AMS were used to prove the presence of sulfur-containing organics.

**Revision in the manuscript:**

**Line 276, Delete:** "According to the linear fitting between the concentration of formed $SO_4^{2-}$ and the amount of consumed $SO_2$ (after wall loss correction for $SO_2$, sulfuric acid gas and sulfate), there was a large gap between the slope of the line and the ratio of $M(SO_4^{2-})$ and $M(SO_2)$, as shown in Fig. S8. There are some possible reasons for this, including the underestimation of deposition and heterogeneous reaction of sulfur species on the wall, the formation of organic sulfur-containing products, and small leaks of pollutants from the smog chamber."

**Comment 16:** Line 282: Too many "This phenomenon indicated".

**Response:** Thanks very much for your comment. The sentences involving "This phenomenon indicated" have been modified in the revised manuscript.

**Revision in the manuscript:**

**Line 336, Change** "This phenomenon further indicated that $NH_4NO_3$ became a dominant nitrate species in the presence of $NH_3$." **to:** "Therefore, $NH_4NO_3$ was the dominant nitrate species in the presence of $NH_3$."

**Line 341, Change** "These phenomena indicated" **to:** ", which revealed"

**Lines 435-436, Change** "This phenomenon indicated that the formation of more-oxygenated organic aerosol and nitrogen-containing organics was enhanced with the increase of $NH_3$ concentration." **to:** "Thence, the formation of more-oxygenated

organic aerosol and nitrogen-containing organics will be enhanced with the increase of

$NH_3$ concentration."

**Comment 17:** Line 286: Too many "This result indicated that".

**Response:** Thanks very much. The sentences involving "This result indicated that" have been deleted and modified in the revised manuscript.

**Revision in the manuscript:**

**Line 339, Delete** "This result indicated that $NH_3$ was incorporated in the photo-oxidation of gasoline vapor.". Similar description has been expressed in the above sentence.

**Line 467, Delete** "Previous studies have indicated that".

**Comment 18:** Line 402-403: Repeated conclusion.

**Response:** Thanks for your comment. This repeated sentence has been deleted in the revised manuscript.

**Revision in the manuscript:**

**Line 451, Delete:** "Our results indicate that the photo-oxidation of gasoline/$NO_x$ in the presence of $SO_2$ and $NH_3$ is a significant source of SA. Therefore, in order to mitigate $PM_{2.5}$ pollution in China, emission control strategies should not only pay attention to primary particulate emissions, but also focus on synergistic reduction of the emission of SA precursors including $NO_x$, $SO_2$, $NH_3$ and, particularly, VOCs."

**Comment 19:** Line 422-423: Repeated conclusion.

**Response:** Thank you very much for your valuable comment. This repeated sentence has been deleted and modified in the revised manuscript.

**Revision in the manuscript:**

**Line 471, Delete:** "Therefore, more attention should be paid to collaborative control reductions in vehicular evaporative emissions and gaseous pollutants, including $NO_x$, $SO_2$, and $NH_3$."

**Lines 471-475, Change** "This will contribute to reducing the burden of $PM_{2.5}$, and then cut the environmental, economic and health costs caused by PM pollution. Corresponding emission controls should be taken into consideration by policy makers for future management." **to:** "Therefore, under the compound pollution conditions of $SO_2$ and $NH_3$, synergistic emission reduction of vehicular evaporative emissions, $SO_2$ (e.g., coal-fired flue gas) and $NH_3$ (e.g., emitted from agricultural non-point source and traffic emissions) should be taken into consideration by policy makers for future management, which will contribute to reducing the burden of $PM_{2.5}$, and then cut the environmental, economic and health costs caused by PM pollution."

**Comment 20:** Line 410-412: SA formation still be a small part from vehicular evaporation emissions. What is the meaning of your study?

**Response:** Thanks for your comment. First, an updated emission inventory of vehicular evaporative emissions have demonstrated that they have become a non-negligible contributor (39.20 %) to ambient VOCs from anthropogenic sources compared with

vehicular tailpipe emissions (Liu et al., 2017). In the past two decades, policy makers and researchers have mainly focused on tailpipe emissions but paid little attention to evaporative emissions from vehicles in China. For example, the evaporative emission limits in the China V standards, which was implemented nationally in 2018, are not much different from the limits in the China I standards implemented in 2000. Our results indicates that the photo-oxidation of gasoline vapor especially in the presence of $SO_2$ and $NH_3$ have a huge secondary aerosol formation potential. When the lower aromatics content (~ 10%) in vehicular evaporative emissions was considered (Zhang et al., 2013), the SA yield is roughly estimated to be about 0.20 and the SA production is roughly estimated to be 0.33 Tg yr$^{-1}$, which is about 1.5 times as much as the primary $PM_{2.5}$ emissions from transportation (0.21 Tg yr$^{-1}$) in China (Jing et al., 2015; Zhang et al., 2007) and accounting for about 21 % of the SOA production (1.6 Tg yr$^{-1}$) from anthropogenic precursors estimated by global chemical transport model (Farina et al., 2010). Thence, under the compound pollution conditions of $SO_2$ (e.g., coal-fired flue gas) and $NH_3$ (e.g., emitted from agricultural non-point source and traffic emissions), the contribution of vehicle evaporative emissions to SOA formation cannot be ignored. Secondary, our results have revealed that vehicular evaporative emissions is a potential source of sulfur- and nitrogen-containing organics in the presence of $NO_x$, $SO_2$ and $NH_3$. Sulfur- and nitrogen-containing organics will have an adverse influence on the climate by light absorption and/or by affecting aerosol hygroscopicity (Staudt et al., 2014; Nguyen et al., 2012), and they also have a significant contribution to SOA and nitrogen or sulfur budgets (Lee et al., 2016; Shang et al., 2016).

Therefore, more attention should be paid to this primary emissions source (i.e., vehicular evaporative emissions), and especially its associated secondary aerosol formation, which is also the significance of our study. According to our study, synergistic emission reduction of vehicular evaporative emissions, $SO_2$ (e.g., coal-fired flue gas) and $NH_3$ (e.g., emitted from agricultural non-point source and traffic emissions) should be taken into consideration by policy makers for future management, which will contribute to reducing the burden of $PM_{2.5}$, and then cut the environmental, economic and health costs caused by PM pollution.

Additionally, we agree that there might be some differences between the VOCs composition of gasoline vapors directly injected to the smog chamber and vehicular evaporative emissions. Thus, further work should be focused on SA formation directly from vehicular evaporative emissions to shed light on the formation mechanism of SA under more atmospherically relevant conditions.

**Revision in the manuscript:**

[revised manuscript text omitted]

---

## Author Comment (AC2) · 27 Apr 2019

**Responses to Referee #2's comments**

**General comment:** Chen et al. propose a study presenting the impact of $NH_3$ and $SO_2$ on secondary organic aerosol formation produced from the oxidation of unburned gasoline vapors. While this study presents interesting results, some aspects (see comments below) remain unclear and/or should be better discussed. One of my main comment is the relevance of the mix of VOC studied in this work when the authors claimed to study relevant unburned gasoline vapors. Indeed, the emission inventory proposed by Liu et al appears very different than the mix of VOC employed in this work. Overall this can lead to an overestimation of the importance of this potential SOA source.

**Response:** Many thanks for your constructive comments and valuable suggestions, which would be much helpful to improve the scientific merits of this manuscript. The concerns raised by you have been carefully addressed in the revised manuscript.

**Response to your main comment:** In our study, the utilized gasoline vapors were also dominated by alkanes (C6 to C12, such as methylcyclopentane ($C_6H_{12}$) and methylcyclohexane ($C_7H_{14}$)), which made up a 65.1 % share. These alkanes were also detected by Liu et al., 2017 and previous studies (Liu et al., 2008; Zhang et al., 2013), and also accounting for a certain proportion in evaporative emission. As for those short chain alkanes (e.g., i-pentane, n-butane and i-butane) reported by Liu et al., 2017, they should also be in our gasoline vapors according to their relative high vapor pressure, while they could not be detected by our GC-MS system due to the separation capacity of the GC column.

Previous studies have reported that long-chain (C6 to C19) alkanes, which are intermediate volatility organic compounds (IVOCs) (Donahue et al., 2006), could also contribute to SOA formation (Pye and Pouliot, 2012; Tkacik et al., 2012; Presto et al., 2009; Lim and Ziemann, 2005; Zhao et al., 2016). In order to relatively accurately predict the contribution of vehicle evaporative emissions to secondary aerosols, the lower aromatics content (~ 10%) and long-chain alkanes in vehicular evaporative emissions was considered (Zhang et al., 2013) and discussed in the revised manuscript. Additionally, we agree that there might be some differences between the VOCs composition of gasoline vapors directly injected to the smog chamber and vehicular evaporative emissions. Thus, further work should be focused on SA formation directly from vehicular evaporative emissions to shed light on the formation mechanism of SA under more atmospherically relevant conditions.

**Revision in the manuscript:**

**Lines 91-95, Add:** "In addition to short-chain alkanes, a certain proportion of aromatics and alkanes (C6 to C12) were also contained in the evaporative emissions (Liu et al., 2008; Zhang et al., 2013). Previous studies have reported that aromatics and long-chain (C6 to C19) alkanes, which are intermediate volatility organic compounds (IVOCs) (Donahue et al., 2006), could contribute to SOA formation (Pye and Pouliot, 2012; Tkacik et al., 2012; Lim and Ziemann, 2005)."

**Line 109, Add:** "65.1 % (v/v) alkanes (C6 to C12),"

**Line 206, Add:** ", methylcyclopentane, methylcyclohexane"

**Lines 452-460, Change** "Considering the typical concentrations of $SO_2$ and $NH_3$ of 40

ppb and 23 ppb in haze pollution in the north China plain (Cheng et al., 2016), the SA yield is roughly estimated to be about 0.3. Then, the SA formed from the photo-oxidation of VOCs emitted by vehicular evaporation in the presence of $SO_2$ and $NH_3$ is roughly estimated to be 0.49 Tg $yr^{-1}$, which is about twice as much as the primary $PM_{2.5}$ emissions from transportation (0.21 Tg in 2007) in China (Jing et al., 2015; Zhang et al., 2007)." **To** "Considering the typical concentrations of $SO_2$ and $NH_3$ of 40 ppb and 23 ppb in haze pollution in the north China plain (Cheng et al., 2016), and the lower aromatics content (~ 10%) in vehicular evaporative emissions (Zhang et al., 2013), the SA yield is roughly estimated to be about 0.20. Recently, an updated emission inventory of vehicular evaporative emissions was reported to be 1.65 Tg $yr^{-1}$ (Liu et al., 2017a). Then, the SA formed from the photo-oxidation of VOCs emitted by vehicular evaporation in the presence of $SO_2$ and $NH_3$ is roughly estimated to be 0.33 Tg $yr^{-1}$, which is about 1.5 times as much as the primary $PM_{2.5}$ emissions from transportation (0.21 Tg $yr^{-1}$) in China (Jing et al., 2015; Zhang et al., 2007) and accounting for about 21 % of the SOA production (1.6 Tg $yr^{-1}$) from anthropogenic precursors estimated by global chemical transport model (Farina et al., 2010)."

**Lines 476-480, Add:** "Additionally, there might be some differences between the VOCs composition of gasoline vapors directly injected to the smog chamber and vehicular evaporative emissions. Thus, further work should be focused on SA formation directly from vehicular evaporative emissions under coexisting $SO_2$ and $NH_3$ conditions to shed light on the formation mechanism of SA under more atmospherically relevant conditions."

**Comment 1:** Lines 49-53: If the highest concentrations of NH$_3$ were observed in urban areas, is the main source from agriculture or vehicle emissions?

**Response:** Thank you very much. This concentrations of NH$_3$ were reported by Ianniello et al. (2010), who carried out the intensive measurements in the campus of Peking University (PKU), located at North of Beijing (39°59′23″ N, 116°18′19″ E), China. And the highest NH$_3$ concentrations in summer were mainly from agricultural activity and fertilizer use, which were regionally transported from south and southeast of Beijing, while could not exclude the influence by traffic emissions at local Beijing. The moderate correlations were obtained between NH$_3$ and pollutants mainly emitted by motor-vehicle exhausts, such as NO$_x$, and CO, indicating an influence by traffic emissions. Recently studies have reported that the contribution of traffic emissions to NH$_3$ had an ascend trend accompanied the increasing urban population and the introduction of vehicles three-way catalytic converters (Pan et al., 2016; Kang et al., 2016).

**Revision in the manuscript:**

**Lines 54-56, Add** ", which mainly derived from the regionally transportation of agricultural activity and fertilizer use, while could not exclude the influence by traffic emissions at local Beijing (Pan et al., 2016; Kang et al., 2016)."

**Comment 2:** Lines 66-70: Split the references and cite proper references. For instance, the work from Kamens and al. should be better acknowledged.

**Response:** Thanks for your valuable suggestion. The references have been split, and

the proper references have been cited in the revised manuscript.

**Revision in the manuscript:**

**Lines 70-71, Add** "Jang and Kamens (2001) first reported the acid-catalytical effect of acidic $H_2SO_4$ on the oxidation of atmospheric carbonyls."

**Lines 71-76, Change** "It has been found that $SO_2$ promotes SA formation from typical biogenic (e.g., isoprene and α-pinene) and anthropogenic (e.g., toluene, o-xylene, 1,3,5-trimethylbenzene, and gasoline vehicle exhaust) precursors through acid-catalyzed reactions (Chu et al., 2016; Liu et al., 2016; Lin et al., 2013; Santiago et al., 2012; Jaoui et al., 2008; Kleindienst et al., 2006; Edney et al., 2005)" **to:** "And the promotion effect of $SO_2$ were further found on the SA formation from typical biogenic (e.g., isoprene and α-pinene) (Lin et al., 2013; Jaoui et al., 2008; Kleindienst et al., 2006; Edney et al., 2005) and anthropogenic (e.g., toluene, o-xylene, 1,3,5-trimethylbenzene, and gasoline vehicle exhaust) precursors (Chu et al., 2016; Liu et al., 2016; Santiago et al., 2012) through acid-catalyzed heterogeneous reactions (Jang et al., 2002; Jang et al., 2003a, b; Czoschke et al., 2003)"

**Comment 3:** Lines 83-86: The authors should better discuss this section and not overestimate the potential of VTE in producing aerosols. According to Liu et al., 2017, the VOC emitted from the evaporation are mainly dominated by short chain alkane (e.g., Pentane, butane,...) which have a very limited impact on SOA formation. However, the authors mainly discussed/focused on the oxidation of aromatics. In addition, important papers are missing and should be discussed; e.g., Pye and Pouliot 2012, Tkacik et al.,

2012.

**Response:** Thanks for your valuable suggestion. In our study, the utilized gasoline vapors were also dominated by alkanes (C6 to C12), which made up a 65.1 % share. These alkanes were also detected by Liu et al., 2017 and previous studies (Liu et al., 2008; Zhang et al., 2013), and also accounting for a certain proportion in evaporative emission. The time variations of long-chain alkanes (methylcyclopentane and methylcyclohexane) in photo-oxidation of gasoline/$NO_x$ in the presence or absence of $SO_2$ and $NH_3$ have been added in Fig. S3 in the Supplement (shown in Fig. R1). As for those short chain alkanes (e.g., i-pentane, n-butane and i-butane) reported by Liu et al., 2017, they should also be in our gasoline vapors according to their relative high vapor pressure, while they could not be detected by our GC-MS system due to the separation capacity of the GC column.

Previous studies have reported that long-chain (C6 to C19) alkanes, which are intermediate volatility organic compounds (IVOCs) (Donahue et al., 2006), could also contribute to SOA formation (Pye and Pouliot, 2012; Tkacik et al., 2012; Presto et al., 2009; Lim and Ziemann, 2005; Zhao et al., 2016). In order to relatively accurately predict the contribution of vehicle evaporative emissions to secondary aerosols, the lower aromatics content (~ 10%) and long-chain alkanes in vehicular evaporative emissions was considered (Zhang et al., 2013) and discussed in the revised manuscript. Additionally, we agree that there might be some differences between the VOCs composition of gasoline vapors directly injected to the smog chamber and vehicular evaporative emissions. Thus, further work should be focused on SA formation directly

from vehicular evaporative emissions to shed light on the formation mechanism of SA under more atmospherically relevant conditions.

**Revision in the manuscript:**

**Lines 91-95, Add:** "In addition to short-chain alkanes, a certain proportion of aromatics and alkanes (C6 to C12) were also contained in the evaporative emissions (Liu et al., 2008; Zhang et al., 2013). Previous studies have reported that aromatics and long-chain (C6 to C19) alkanes, which are intermediate volatility organic compounds (IVOCs) (Donahue et al., 2006), could contribute to SOA formation (Pye and Pouliot, 2012; Tkacik et al., 2012; Lim and Ziemann, 2005)."

**Line 109, Add:** "65.1 % (v/v) alkanes (C6 to C12),"

**Line 206, Add:** ", methylcyclopentane, methylcyclohexane"

**Lines 452-460, Change** "Considering the typical concentrations of $SO_2$ and $NH_3$ of 40 ppb and 23 ppb in haze pollution in the north China plain (Cheng et al., 2016), the SA yield is roughly estimated to be about 0.3. Then, the SA formed from the photo-oxidation of VOCs emitted by vehicular evaporation in the presence of $SO_2$ and $NH_3$ is roughly estimated to be 0.49 Tg yr$^{-1}$, which is about twice as much as the primary $PM_{2.5}$ emissions from transportation (0.21 Tg in 2007) in China (Jing et al., 2015; Zhang et al., 2007)." **To** "Considering the typical concentrations of $SO_2$ and $NH_3$ of 40 ppb and 23 ppb in haze pollution in the north China plain (Cheng et al., 2016), and the lower aromatics content (~ 10%) in vehicular evaporative emissions (Zhang et al., 2013), the SA yield is roughly estimated to be about 0.20. Recently, an updated emission inventory of vehicular evaporative emissions was reported to be 1.65 Tg yr$^{-1}$ (Liu et al., 2017a).

Then, the SA formed from the photo-oxidation of VOCs emitted by vehicular evaporation in the presence of $SO_2$ and $NH_3$ is roughly estimated to be 0.33 Tg $yr^{-1}$, which is about 1.5 times as much as the primary $PM_{2.5}$ emissions from transportation (0.21 Tg $yr^{-1}$) in China (Jing et al., 2015; Zhang et al., 2007) and accounting for about 21 % of the SOA production (1.6 Tg $yr^{-1}$) from anthropogenic precursors estimated by global chemical transport model (Farina et al., 2010)."

**Lines 460-462, Add:** "In addition, the photo-oxidation of long-chain alkanes (> C6, IVOCs) contained in evaporative emissions would also contribute to SOA formation (Pye and Pouliot, 2012; Tkacik et al., 2012; Presto et al., 2009; Lim and Ziemann, 2005; Zhao et al., 2016)."

**Lines 476-480, Add:** "Additionally, there might be some differences between the VOCs composition of gasoline vapors directly injected to the smog chamber and vehicular evaporative emissions. Thus, further work should be focused on SA formation directly from vehicular evaporative emissions under coexisting $SO_2$ and $NH_3$ conditions to shed light on the formation mechanism of SA under more atmospherically relevant conditions."

Fig. R1 has been added in the Supplement.

[Figure]

Fig. R1. Time variations of organic gas-phase species (a) Benzene, (b) Toluene, (c) C2-Benzene, (d) C3-Benzene, (e) C4-Benzene, (f) Methylcyclopentane, and (g) Methylcyclohexane in photo-oxidation of gasoline/$NO_x$ in the presence or absence of $SO_2$ and $NH_3$. Letters in abbreviations represent the reactants introduced into the chamber reactor, i.e., "G" represents gasoline, "N" represents nitrogen oxides, "S" represents sulfur dioxide, "A" represents ammonia.

**Comment 4:** Lines 121-122: Please change to the widely used acronym (PTR-TOF), (i.e., Yuan et al., 2017 Chem Review).

**Response:** Thanks for your suggestion. HR-ToF-PTRMS has been changed to PTR-TOF in the revised manuscript.

**Revision in the manuscript:**

**Lines 131-132, Change** "high-resolution time-of-flight proton transfer reaction mass spectrometry (HR-ToF-PTRMS)" **To** "proton-transfer-reaction time of flight mass spectrometry (PTR-TOF)"

**Line 133, Add reference "**Yuan et al., 2017**"**

**Line 207, Change** "HR-ToF-PTRMS" **To** "PTR-TOF"

**Comment 5:** Lines 130-131: Is the density comparable to other studies?

**Response:** Thanks for your comment. Many previous studies have demonstrated that PM effective density could be estimated by comparing mobility diameter from the DMA (i.e., $d_m$) and vacuum aerodynamic diameter from an Aerodyne AMS (i.e., $d_{va}$) in parallel, i.e., $\rho = d_{va}/d_m$ (Jimenez et al., 2003a, b; DeCarlo et al., 2004; Bahreini et al., 2005; Alfarra et al., 2006; Ng et al., 2007; Sato et al., 2010). According to this equation, the density of PM was calculated to be $1.5-1.6$ g cm$^{-3}$, which was in the range of density of SOA derived from aromatic hydrocarbons ($1.24-1.48$ g cm$^{-3}$) (Sato et al., 2010) and ammonium nitrate ($NH_4NO_3$, $1.72$ g cm$^{-3}$) (Bahreini et al., 2005) and could be comparable with the previous studies (Li et al., 2018).

**Revision in the manuscript:**

**Lines 141-143, Add** ", which was in the range of density of SOA derived from aromatic hydrocarbons (1.24−1.48 g cm$^{-3}$) (Sato et al., 2010) and ammonium nitrate (NH$_4$NO$_3$, 1.72 g cm$^{-3}$) (Bahreini et al., 2005) and could be comparable with the previous studies (Li et al., 2018)"

**Comment 6:** Lines 191-192: The authors should be careful here. The PTR can measure a certain (small) subset of OVOC produced from a given reaction. Therefore, it is not because the analytical technique was not able to identify any significant differences that mean SO$_2$/NH$_3$ did not significantly impact the gaseous phase. As an example, a large formation of H$_2$SO$_4$ is mentioned in the paper. In addition, the authors should propose a more quantitative comparison (as realized with the AMS data) and not only briefly compare the mass spectra.

**Response:** Thanks for your valuable suggestion. The gas-phase intermediates, such as small molecule oxygenated VOCs (OVOC) (e.g., acetic acid) formed during the photo-oxidation of gasoline/NO$_x$ under different conditions have been discussed in the revised manuscript and added in the Supplement (shown in Fig. R2). The time series of acetic acid concentration measured by PTR-TOF showed a decreased trend in the presence of SO$_2$, suggesting that the uptake of acetic acid might be enhanced. This phenomenon was consistent with those reported by Liggio and Li (2006), who observed that the uptake of organic compounds under acidic conditions would be enhanced significantly. Moreover, the presence of high concentrations of SO$_2$ would generate gaseous H$_2$SO$_4$, which would contribute to the formation of particle phase. Similarly, the concentration

of acetic acid also shown an obviously decreased trend in the presence of $NH_3$, which might be caused by the reaction of acid-base reaction or the uptake of acetic acid in the presence of $NH_3$ (Liu et al., 2015).

**Revision in the manuscript:**

**Lines 214-224, Add:** "However, as for the gas-phase intermediates formed during the photo-oxidation of gasoline/$NO_x$ under different conditions, such as small molecule oxygenated VOCs (OVOCs), which could also be measured by PTR-TOF. The time series of OVOCs concentration would vary with the concentration of $SO_2$ and $NH_3$. For example, we observed that acetic acid concentration decreased with the increased concentration of $SO_2$ (Fig. S5, in the Supplement), suggesting that the uptake of acetic acid may be enhanced. This phenomenon was consistent with those reported by Liggio and Li (2006), who observed that the uptake of organic compounds under acidic conditions would be enhanced significantly. Moreover, the presence of high concentrations of $SO_2$ would generate gaseous $H_2SO_4$, which would contribute to the formation of particle phase, as discussed in the next section. Similarly, the concentration of acetic acid also shown an obviously decreased trend in the presence of $NH_3$ (Fig. S5, in the Supplement), which could be caused by the reaction of acid-base reaction or the uptake of acetic acid in the presence of $NH_3$ (Liu et al., 2015c)."

Fig. R2 has been added in the Supplement.

[Figure]

Fig. R2. Time variations of acetic acid during the photo-oxidation of gasoline/NO$_x$ in the presence or absence of SO$_2$ and NH$_3$. Letters in abbreviations represent the reactants introduced into the chamber reactor, i.e., "G" represents gasoline, "N" represents nitrogen oxides, "S" represents sulfur dioxide, "A" represents ammonia.

**Line 210, Delete:** "In addition, the typical mass spectra of organic gas-phase species derived from HR-ToF-PTRMS after 480 min of the photo-oxidation reaction at different concentrations of SO$_2$ or NH$_3$ are shown in Fig. S5, and no significant differences were found. Therefore, it is reasonable to deduce that the presence of SO$_2$ or NH$_3$ did not significantly impact the initial gas-phase oxidation mechanism of gasoline. This was consistent with the previous study conducted by Chu et al. (2016), who found that the presence of SO$_2$ and NH$_3$ did not significantly impact the initial gas-phase oxidation of toluene in the presence of NO$_x$."

**Comment 7:** Line 206: The background of ammonia appears quite subsequent. Do the authors have an explanation? Is it an expected background in chamber experiments?

**Response:** Thanks for your comment. The background of NH$_3$ might be present in the

background air, it also might be introduced during the humidification process of the smog chamber. Similar phenomenon was reported by Liu et al. (2015), who found that the concentration of background $NH_3$ in the dry chamber (9 $m^3$ chamber at Environment Canada) was consistently at ~5 ppb (after cleaning), while increasing to a reproducible ~35 ppb after humidifying to 50 % RH.

The background $NH_3$ might also be derived from the partitioning of the deposited ammonium sulfate and ammonium nitrate on the chamber wall when humid air was introduced. Unfortunately, appropriate instruments are unavailable to measure the exact concentration of $NH_3$ in the background air in the chamber.

**Revision in the manuscript:**

**Lines 322-323, Add** "or introduced during the humidification process of the chamber (Liu et al., 2015c)"

**Lines 323-324, Add** "Unfortunately, appropriate instruments are unavailable to measure the exact concentration of background $NH_3$ in the chamber."

**Comment 8:** Fig.3 the color scale is hard to read please modify it.

**Response:** Thanks for your suggestion. The color scale and the font size in Fig. 3 has been modified in the revised manuscript, as shown in Fig. R3.

**Revision in the manuscript:**

Fig. R3 has been modified in the revised manuscript.

[Figure]

Fig. R3. Time series of the size distributions for the generated secondary aerosol during the photo-oxidation experiments with different $SO_2$ initial concentrations (Exps. GN, SGN1, SGN2, SGN3, and SGN4). $D_{p,max}$ and $N_{max}$ represent the maximal diameter and number concentration of generated secondary aerosol, respectively, during each photo-oxidation experiment.

**Comment 9:** Lines 213-219: In this section, there are some inconsistencies with the literature. The number (i.e., up to 50 times more particles) reported in earlier studies are different than in the work proposed by Chen et al. However the concentration of $SO_2$ and the precursors were similar. For example, while the $SO_2$ concentration was multiplied by 4 the formation of particles didn't increase significantly. The authors should explain such discrepancy and discuss/compare their results to the existing literature.

**Response:** Thanks for your valuable suggestion. The higher magnification (up to 50 times) of $SO_2$ reported by Liu et al. (2016) might be related to the different precursor

systems (gasoline vehicle exhaust), higher initial mixing ratios of precursors (2.2−4.3 ppm) and higher concentration of $NO_x$ (300−450 ppb) (Liu et al., 2016). There were difference in the VOCs composition between evaporative emissions and gasoline vehicle exhaust, especially the aromatic and IVOCs (Liu et al., 2017). Our recent study demonstrated that SOA formation could be significantly enhanced by the increase of aromatic content (Chen et al., 2019). Previous studies have also demonstrated that those unspeciated organic emissions (e.g., IVOCs) from gasoline vehicle exhaust have a significant contribution to SOA formation (Jathar et al., 2014; Gordon et al., 2014). Meanwhile, for the gasoline vehicle exhaust reaction systems reported by Liu et al. (2016), a small amount of POA was also present in the initial reaction systems. These enhanced SOA formation and the pre-existing POA would provide larger surface areas for the condensation and heterogeneous uptake of low-volatility vapors (e.g., gaseous $H_2SO_4$), then promoting a higher magnification in particle number concentrations in the presence of $SO_2$.

As for the discrepancy between the magnification of particle number concentrations and $SO_2$ concentrations, on one hand, there might be some particles, especially nanoclusters, were lost to the chamber wall and not be detected; on the other hand, the gaseous $H_2SO_4$ generated from the presence of high concentrations of $SO_2$ would contribute to the nanoclusters formation (Chu et al., 2019; Sipilä et al., 2010), and then grow to sizes large enough be detected. While the initial size of nanoclusters (sub-3 nm) might be too small to be detected by our general SMPS. That is to say, the particle number concentrations measured by our SMPS might be the particles after

growing up by collision. This could be supported by the enhancement in the particle diameters (144−172 nm) and sulfate concentrations (13−38 μg m$^{-3}$) with the increased concentration of SO$_2$. After considering the underestimation of particles formation (factor of 1.97−2.82), which was evaluated according to the methods described by Zhang et al. (2014) (seen in Section 2.3), the sulfate concentrations will be enhanced by a factor of 5.8 when comparing between experiments SGN 1 and SGN 4.

**Revision in the manuscript:**

**Lines 244-254, Add:** "This higher magnification of SO$_2$ might be related to the different VOCs composition between evaporative emissions and gasoline vehicle exhaust, especially the aromatic and IVOCs (Liu et al., 2017). Our recent study demonstrated that SOA formation could be significantly enhanced by the increase of aromatic content (Chen et al., 2019b). Those unspeciated organic emissions (e.g., IVOCs) from gasoline vehicle exhaust would also have a significant contribution to SOA formation (Jathar et al., 2014; Gordon et al., 2014). Moreover, a small amount of POA was present in the initial reaction systems in Liu et al. (2016). These enhanced SOA formation and the pre-existing POA would provide larger surface areas for the condensation and heterogeneous uptake of low-volatility vapors (e.g., gaseous H$_2$SO$_4$), then promoting a higher magnification in particle number concentrations in the presence of SO$_2$. The higher initial mixing ratios of precursors (2.2−4.3 ppm) was also present in the reaction systems conducted by Liu et al. (2016), which would further be beneficial to the SOA formation."

**Lines 266-276, Add:** "Additionally, it is worth noting that there was a discrepancy

between the magnification of particle number concentrations, surface areas and $SO_2$ concentrations. On one hand, there might be some particles, especially nanoclusters, were lost to the chamber wall and not be detected; on the other hand, the initial size of nanoclusters contributed from gaseous $H_2SO_4$ was small enough (sub-3 nm) (Chu et al., 2019; Sipilä et al., 2010) and couldn't be detected by our general SMPS. That is to say, the particle number concentrations and surface areas measured by our SMPS might be the particles after growing up by collision. This could be supported by the enhancement in the particle diameters (144−172 nm) and sulfate concentrations (13−38 μg m$^{-3}$) in the presence of $SO_2$. After considering the underestimation of particles formation (factor of 1.97−2.82, seen in Section 2.3), the sulfate concentrations will be enhanced by a factor of 5.8 when comparing between experiments SGN 1 and SGN 4."

**Comment 10:** Lines 227-232: This is in line with my previous comments. How do they explain such a moderate increase between SGN 1 and 4?

**Response:** Thanks for your valuable suggestion. The gaseous $H_2SO_4$ generated from the presence of high concentrations of $SO_2$ would contribute to the nanoclusters formation (Chu et al., 2019; Sipilä et al., 2010), and then grow to sizes large enough be detected. While the initial size of nanoclusters (sub-3 nm) might be too small to be detected by our general SMPS. That is to say, the particle surface areas measured by our SMPS might be the particles after growing up by collision. This could be supported by the enhancement in the particle diameters (144−172 nm) and sulfate concentrations (13−38 μg m$^{-3}$) in the presence of $SO_2$. After considering the underestimation of

particles formation (factor of 1.97−2.82), which was evaluated according to the methods described by Zhang et al. (2014) (seen in Section 2.3), the sulfate concentrations will be enhanced by a factor of 5.8 when comparing between experiments SGN 1 and SGN 4.

**Revision in the manuscript:**

**Lines 266-276, Add:** "Additionally, it is worth noting that there was a discrepancy between the magnification of particle number concentrations, surface areas and $SO_2$ concentrations On one hand, there might be some particles, especially nanoclusters, were lost to the chamber wall and not be detected; on the other hand, the initial size of nanoclusters contributed from gaseous $H_2SO_4$ was small enough (sub-3 nm) (Chu et al., 2019; Sipilä et al., 2010) and couldn't be detected by our general SMPS. That is to say, the particle number concentrations and surface areas measured by our SMPS might be the particles after growing up by collision. This could be supported by the enhancement in the particle diameters (144−172 nm) and sulfate concentrations (13−38 μg m$^{-3}$) in the presence of $SO_2$. After considering the underestimation of particles formation (factor of 1.97−2.82, seen in Section 2.3), the sulfate concentrations will be enhanced by a factor of 5.8 when comparing between experiments SGN 1 and SGN 4."

**Comment 11:** Lines 245-246: Which compounds are the authors referring to? Products from heterogeneous reactions or Aldehydes? In both cases, the sentence should be revised. (i) products formed in the particle phase cannot partition as they are already in the condensed phase and second the aldehydes are not considered as low vapor pressure

compounds (Kroll and Seinfeld, 2008).

**Response:** Thanks for your valuable suggestion. These products referred to products generated through the acid-catalyzed heterogeneous reactions. The sentences in Lines 245−246 have been modified in the revised manuscript.

**Revision in the manuscript:**

**Lines 288-289, Change:** "These low-vapor-pressure products preferentially partition into the particle phase and subsequently contribute to the SOA formation" to "These low-vapor-pressure products generated from heterogeneous reactions preferentially contribute to the SOA formation"

**Line 289, Add reference:** "Kroll and Seinfeld, 2008"

**Comment 12:** Lines 249-253: Before speculating on the potential formation of organosulfur the authors should present/discuss the validation of the AMS results: calibration of the AMS, uncertainties of the measurements, SMPS vs AMS data,... This is important in order to claim a large formation of organosulfur.

**Response:** Thanks for your valuable suggestion. More details about the validation of the HR-ToF-AMS results has been added in the Materials and Methods section.

**Revision in the manuscript:**

**Lines 145-153, Add:** "For all experiments, the HR-ToF-AMS operated in a cycle including two modes, 3 min V mode and 2 min W mode. Specifically, V mode (higher signal) can obtain the mass concentrations of the aerosols and W mode (higher resolution) can obtain high resolution mass spectral data. The inlet flow rate, ionization

efficiency (IE), and particle sizing were calibrated according to the standard protocols (Drewnick et al., 2005; Jimenez et al., 2003c; Jayne et al., 2000), using the size-selected pure ammonium nitrate (AN) particles. All HR-ToF-AMS data were analyzed with ToF-AMS analysis toolkit SQUIRREL 1.57I/PIKA 1.16I version, in Igor Pro Version 6.37. HR-ToF-AMS results were also corrected using the mass concentration derived from SMPS according to the same method as Gordon et al. (2014), details of this correction are shown in the Supplement."

**Supplement,** details of the HR-ToF-AMS correction have been added as follows:

**S1. AMS Corrections: Comparison with SMPS Measurements**

Theoretically, the sum of the secondary aerosol (SA) mass measured by HR-ToF-AMS should be equal to the mass calculated from the SMPS size distributions. However, both methods have their limitations, in which SMPS measures particle mobility diameter, while HR-ToF-AMS measures mass. Therefore, particle shape and density must be assumed before converting SMPS measurements to mass. Here, we assume that particles are spherical, and the density of SA were calculated from the equation $\rho = d_{va}/d_m$, where $d_{va}$ is the mean vacuum aerodynamic diameter measured by an HR-ToF-AMS and $d_m$ is the mean electrical mobility diameter measured by SMPS (DeCarlo et al., 2004). However, fractal-like particles will cause the SMPS to overestimate the spherical equivalent diameter and therefore overestimate the particle mass. While, HR-ToF-AMS tends to underestimate the SA mass due to the transmission efficiency (Liu et al., 2007) and collection efficiency (Takegawa et al., 2005).

For all the experiments with the discrepancies between HR-ToF-AMS and SMPS,

we assume that the difference in mass has the same chemical composition as the measured chemical species (i.e., organics, nitrate, sulfate, and ammonium). And then a scaling factor ($AMS_{sf}$) was calculated for each experiment to correct the SA mass measured by HR-ToF-AMS and close the gap with the SMPS measurement. The scaling factor could be calculated as following equation:

$$AMS_{sf} = \frac{C_{SMPS}}{C_{Org} + C_{NO_3} + C_{SO_4} + C_{NH_4}}$$

in which $C_{SMPS}$ is the SA mass concentration derived from SMPS, $C_{Org}$, $C_{NO_3}$, $C_{SO_4}$ and $C_{NH_4}$ are the mass concentrations of organics, nitrate, sulfate, and ammonium measured by HR-ToF-AMS, respectively. The $AMS_{sf}$ for each time step after wall loss correction is calculated and used to scale the AMS data for the entire experiment. For all the experiments the average $AMS_{sf}$ ranged from 1.09 to 1.23.

**Comment 13:** Lines 265-270: In other words, the large gap between $SO_2$ and $SO_4$ cannot be explained by the organosulfur. The authors should not speculate something and a few lines after conclude that is not the case. In Fig S8, the difference should be tens of ug m$^{-3}$ while the authors estimated the formation organosulfur of a few tens of ng m-3.

**Response:** Thanks for your valuable suggestion. Indeed, the large gap between the amount of consumed $SO_2$ and concentration of formed $SO_4^{2-}$ cannot be explained by the organosulfur. We think there might be caused by many reasons as follows:

First, the sulfur-containing organics concentration in our study might be underestimated by the HR-ToF-AMS when considering one cannot resolve all the sulfur-containing

fragments that may exist, and some of the sulfur-containing organics might fragment into masses that do not contain sulfur and thus are quantified as organic. Furthermore, the relative ionization efficiency (RIE) for the sulfur-containing organics fragments was assumed to be equivalent to the remainder of the organics (1.3), since a RIE value for sulfur-containing organics is unknown. This may introduce an additional uncertainty to the quantitation of sulfur-containing organics. Therefore, the estimated concentrations of sulfur-containing organics using HR-ToF-AMS were a conservative lower-bound.

Secondary, sulfur species may exist in the form of gaseous sulfuric acid ($H_2SO_4$) and could not be detected in our laboratory. In fact, it can be measured using a chemical ionization long time-of-flight mass spectrometer (LToF-CIMS, Aerodyne Research, Inc.). Unfortunately, the LToF-CIMS is unavailable in our laboratory. Meanwhile, the deposition and heterogeneous reaction of sulfur species (e.g., sulfate and gaseous sulfuric acid) on the wall might be underestimated in our study.

In order to more directly explain the presence of sulfur-containing organics, we refer to your and the first reviewer's comment (comment 15), the S-bearing organic fragments ($C_xH_yO_zS$) detected by HR-ToF-AMS were used to prove the presence of sulfur-containing organics.

**Revision in the manuscript:**

**Line 292, Delete:** "According to the linear fitting between the concentration of formed $SO_4^{2-}$ and the amount of consumed $SO_2$ (after wall loss correction for $SO_2$, sulfuric acid gas and sulfate), there was a large gap between the slope of the line and the ratio of $M(SO_4^{2-})$ and $M(SO_2)$, as shown in Fig. S8. There are some possible reasons for this,

including the underestimation of deposition and heterogeneous reaction of sulfur species on the wall, the formation of organic sulfur-containing products, and small leaks of pollutants from the smog chamber."

**Supplement, Delete:** Fig. S8.

**Line 307, Add:** "conservative lower-bound"

**Lines 309-316, Add:** "Additionally, it should be noted that the sulfur-containing organics concentration in this study might be underestimated by the HR-ToF-AMS when considering one cannot resolve all the sulfur-containing fragments that may exist, and some of the sulfur-containing organics might fragment into masses that do not contain sulfur and thus are quantified as organic. Furthermore, the relative ionization efficiency (RIE) for the sulfur-containing organics fragments was assumed to be equivalent to the remainder of the organics (1.3), since a RIE value for sulfur-containing organics is unknown. This may introduce an additional uncertainty to the quantitation of sulfur-containing organics."

**Comment 14:** Lines 387-388: The O:C ratio didn't change significantly so I do not think the authors can claim to the formation of highly oxidized species.

**Response:** Thanks for your comment. You are right. Indeed, the O/C didn't change significantly for the SOA generated under the different concentration of $NH_3$. The formation of highly oxidized species is relatively speaking when comparing the Factor 1-N and Factor 2-N. Factor 2-N has a relatively higher O/C (O/C = 0.44) and oxidation state (OSc = -0.42) than Factor 1-N (O/C = 0.32, OSc =-0.57). In order to avoid this

misunderstanding, this description has been corrected in the revised manuscript.

**Revision in the manuscript:**

**Lines 438-439, Change** "highly oxidized species" **To** "more oxidized species"